# Finding separatrices of dynamical flows with Deep Koopman Eigenfunctions

**Kabir V. Dabholkar**
Faculty of Mathematics
Technion – Israel Institute of Technology
Haifa, Israel 3200003
kabir@campus.technion.ac.il

**Omri Barak**
Rappaport Faculty of Medicine and Network Biology Research Laboratory
Technion – Israel Institute of Technology
Haifa, Israel 3200003
omri.barak@gmail.com

## Abstract

Many natural systems, including neural circuits involved in decision making, are modeled as high-dimensional dynamical systems with multiple stable states. While existing analytical tools primarily describe behavior near stable equilibria, characterizing separatrices—the manifolds that delineate boundaries between different basins of attraction—remains challenging, particularly in high-dimensional settings. Here, we introduce a numerical framework leveraging Koopman Theory combined with Deep Neural Networks to effectively characterize separatrices. Specifically, we approximate Koopman Eigenfunctions (KEFs) associated with real positive eigenvalues, which vanish precisely at the separatrices. Utilizing these scalar KEFs, optimization methods efficiently locate separatrices even in complex systems. We demonstrate our approach on synthetic benchmarks, ecological network models, and high-dimensional recurrent neural networks trained on either neuroscience-inspired tasks or fit to real neural data. Moreover, we illustrate the practical utility of our method by designing optimal perturbations that can shift systems across separatrices, enabling predictions relevant to optogenetic stimulation experiments in neuroscience. Our code is available on GitHub and we share an interactive description of the work and its extensions in a UniReps blog.

## 1 Introduction

Recurrent neural networks (RNNs) are widely used in neuroscience as models of computation arising from the coordinated dynamics of many neurons, motivating efforts to reverse-engineer their underlying dynamical mechanisms [1, 2]. In particular, many cognitive tasks such as decision-making [3] and associative memory [4] can be modeled as multistable dynamical systems, where distinct decisions or memories correspond to different stable attractor states in phase space. Transitions between these attractors are governed by the geometry of the basins of attraction and, crucially, by the *separatrix*: the manifold that delineates the boundary between basins (Figure 1A).

A reverse-engineering method that has yielded significant insights about RNN computations involves finding approximate fixed points and linearising around them [5]. This involves minimizing a scalar function—the kinetic energy $q(x) = \|f(x)\|^2$—to locate these points (Figure 1B). Once found, the

39th Conference on Neural Information Processing Systems (NeurIPS 2025).

linearisation of the dynamics at the fixed point can shed light on the mechanism of computations [6–16].

However, fixed points alone do not capture the global organization of multistable dynamics. Since inputs perturb the state in arbitrary directions, it is critical to know whether they cross the separatrix. To predict the effects of perturbations or design targeted interventions, one must characterize the separatrix itself.

Ideally, we would have a scalar function analogous to the kinetic energy—smooth, yet vanishing precisely on the separatrix (Figure 1C). This would allow gradient-based optimization to locate the decision boundary and enable the design of optimal decision-changing perturbations (Figure 1A).

In this work, we propose a novel method to characterize separatrices in high-dimensional black-box dynamical systems by leveraging Koopman operator theory [17, 18]. Specifically, we approximate scalar-valued Koopman eigenfunctions (KEFs) with positive real eigenvalues using deep neural networks. These eigenfunctions vanish precisely on the separatrix.

We apply this framework to synthetic systems, ecological models, RNNs trained on neuroscience-inspired tasks, and trained to reproduce neural recordings. In addition, we demonstrate that the learned KEFs can be used to design minimal perturbations that push the system across separatrices—a setting relevant to experimental protocols such as optogenetic stimulation.

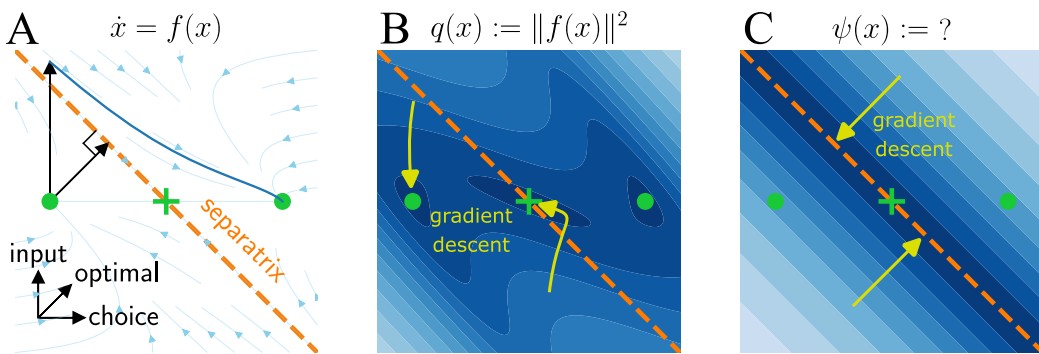

Figure 1: (A) Phase-portrait of a 2D bistable system. The two attractors can signify different choices, and therefore the direction between them is called the choice direction. External input pushes the system across the separatrix letting it relax to the other attractor. The optimal perturbation has a different direction. (B) The kinetic energy vanishes at the fixed points (green 'o' stable and '+' unstable) but does not reveal the full separatrix. (C) We aim to learn a scalar function $\psi(x)$ that vanishes precisely on the separatrix. Gradient descent from random initial points yields a numerical method to locate the respective minima of the scalar functions.

We summarise our main contributions:

- We develop a tool to locate separatrices, the surfaces between basins of attraction in black-box multi-stable dynamical systems: a gap in the RNN reverse-engineering toolkit.

- We demonstrate that KEFs with positive eigenvalues vanish precisely on the separatrix and can be trained using deep neural networks and a loss based on the Koopman PDE error.

- We identify two degeneracies of the Koopman PDE and propose effective regularization strategies to resolve them.

- We show how the learned KEFs can be used to design minimal norm perturbations that shift the system across separatrices.

- We demonstrate the method on systems ranging from low-dimensional models to a 668-dimensional RNN fit to mouse neural data.

## 2   Related Work

Our work builds on a growing body of literature at the intersection of Koopman operator theory, deep learning, and the analysis of dynamical systems, particularly in neuroscience and machine learning.

Koopman theory has recently been used to evaluate similarity between dynamical systems, both in neuroscience [19]—where it is applied to study the temporal structure of computation—and in machine learning, where it has been used to compare training dynamics across models [20]. These approaches typically analyze system-level behavior using dynamic mode decomposition [21–23], a finite-dimensional approximation of the Koopman operator.

In parallel, deep learning methods have emerged as powerful tools for solving partial differential equations (PDEs) in high-dimensions. Notably, the Deep Ritz Method [24] and Deep Galerkin Method (DGM) [25] which eliminate the need for meshes of points. A related line of work uses physics-informed neural networks (PINNs), which incorporate known physics (often PDEs in fluid dynamics) as part of the loss function during DNN training [26].

Koopman-based embeddings have also been proposed as a tool for analyzing the internal dynamics of RNNs. In [27], the authors show that eigenvectors of finite-dimensional approximations of the Koopman operator can uncover task-relevant latent structure in RNNs. More generally, several works explore DNN-based approximations of Koopman operators for learning meaningful embeddings of nonlinear dynamics [28–30].

An alternative line of research for identifying Lagrangian Coherent Structures (LCS) employs the Finite-Time Lyapunov Exponent (FTLE) [31–33], which quantifies sensitivity to initial conditions by measuring the exponential rate of separation between nearby trajectories over a finite time horizon. Ridges in the FTLE field reveal stable and unstable LCS, with the latter corresponding to separatrices in our terminology. These methods are most often applied to two- or three-dimensional fluid flows, where they delineate dynamically distinct regions in the velocity field.

Finally, our approach is conceptually connected to work on the geometry of Koopman eigenfunctions themselves. In particular, [34] studies the level sets of KEFs and their relationship to isostables and isochrons in systems with stable fixed points. In the setting of linear systems, and nonlinear systems topologically conjugate to them, [35, 36] establish theoretical links between KEF level sets and separatrices (stable manifolds). Together, these studies motivate our approach of deep-learning KEFs, as a method for identifying separatrices in general high-dimensional, multi-stable systems.

## 3   Results

### 3.1   KEFs as Scalar Separatrix Indicators

We consider autonomous dynamical systems of the form:

$$\dot{\boldsymbol{x}} = f(\boldsymbol{x}), \quad \boldsymbol{x} \in \mathcal{X} \tag{1}$$

where the $\dot{\Box}$ is shorthand for the time derivative $\frac{d}{dt}\Box$ and $f : \mathcal{X} \to \mathcal{X}$ defines the dynamics on an $N$ dimensional state space $\mathcal{X}$.

Our goal is to construct a smooth scalar function $\psi$ that vanishes precisely on the separatrix between basins of attraction. Consider two such basins (Figure 2). All we care about is the existence of the separatrix manifold, with dynamics moving away from it. The simplest such dynamics is a one-dimensional linear system $\dot{\psi} = \lambda\psi$, with $\lambda > 0$. This motivates a mapping that projects $\boldsymbol{x} \in \mathcal{X}$ to $\psi(\boldsymbol{x}) \in \mathbb{R}$, and induces these dynamics. Such a mapping needs to satisfy:

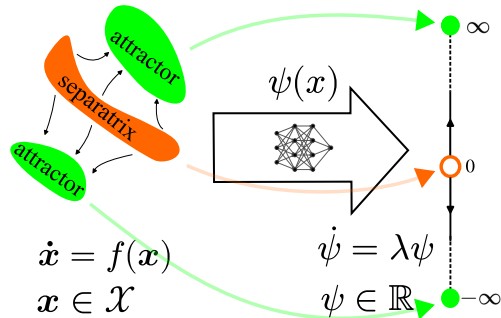

Figure 2: Mapping the high-dimensional dynamics of $\boldsymbol{x}$ with an unstable manifold – the separatrix – to the one dimensional linear dynamics of $\psi$ with instability at $\psi = 0$, i.e., $\lambda > 0$. We approximate the mapping $\psi(\boldsymbol{x})$ with a DNN.

$$\frac{d}{dt}\Big(\psi\big(\boldsymbol{x}(t)\big)\Big) = \lambda\psi\big(\boldsymbol{x}(t)\big) \tag{2}$$

along any trajectory $\boldsymbol{x}(t)$ in $\mathcal{X}$. See Appendix J for a formal link between $\psi$ and the separatrix.

This is precisely the behavior of a Koopman eigenfunction (KEF) with eigenvalue $\lambda > 0$. Note that Koopman eigenfunctions are usually introduced in a different manner, and Appendix C shows the connection to our description.

Equation (2) can be re-written as:

$$\nabla\psi(\boldsymbol{x}) \cdot f(\boldsymbol{x}) = \lambda\psi(\boldsymbol{x}). \tag{3}$$

by employing the chain-rule of differentiation and requiring it to hold at all $\boldsymbol{x} \in \mathcal{X}$. This is the Koopman partial differential equation (PDE). $\lambda$ relates to the timescale of $\psi$ and is an important hyperparameter of our method (Appendix I).

We approximate $\psi$ using a deep neural network (Appendix F) and train it by minimizing the Koopman PDE residual. Specifically, we define the loss:

$$\mathcal{L}_{\text{PDE}} = \mathbb{E}_{\boldsymbol{x}\sim p(\boldsymbol{x})}\left[\nabla\psi(\boldsymbol{x}) \cdot f(\boldsymbol{x}) - \lambda\psi(\boldsymbol{x})\right]^2, \tag{4}$$

where $p(\boldsymbol{x})$ is a sampling distribution over the phase space [24, 25]. As with any eigenvalue problem, this loss admits the trivial solution $\psi \equiv 0$. To discourage such solutions, we introduce a shuffle-normalization loss where the two terms are sampled independently from the same distribution:

$$\mathcal{L}_{\text{shuffle}} = \mathbb{E}_{\boldsymbol{x}\sim p(\boldsymbol{x}),\tilde{\boldsymbol{x}}\sim p(\boldsymbol{x})}\left[\nabla\psi(\boldsymbol{x}) \cdot f(\boldsymbol{x}) - \lambda\psi(\tilde{\boldsymbol{x}})\right]^2, \tag{5}$$

and optimize the ratio:

$$\mathcal{L}_{\text{ratio}} = \frac{\mathcal{L}_{\text{PDE}}}{\mathcal{L}_{\text{shuffle}}}. \tag{6}$$

We train using stochastic gradient descent, where expectations are approximated by a batch of samples drawn from $p(\boldsymbol{x})$ and the shuffle corresponds to a random permutation of the samples in the batch (see Appendix G for details).

To illustrate the method, we start with an analytically solvable system in 1D (Figure 3A):

$$\dot{x} = x - x^3 \tag{7}$$

The system has three fixed points, corresponding to minima of $q(x)$ (Figure 3B). A $\lambda = 1$ KEF can be derived analytically (Appendix A):

$$\psi(x) = \frac{x}{\sqrt{|1 - x^2|}} \tag{8}$$

And the zero of this function corresponds to the unstable point, which serves as a separatrix in this 1D case. Figure 3C shows that the DNN approximates this function well, with the location of the zero (separatrix) being captured precisely.

We also apply the method to two 2D bistable systems: a 2D damped Duffing oscillator (Figure 3DEF), and a 2-unit GRU RNN trained on a one-bit flip-flop task (Figure 3GHI). In both cases, the system has two stable fixed points (green circles) and one unstable saddle (green crosses). Kinetic energy functions, shown for comparison, are minimized at the fixed points. In contrast, the learned $\lambda = 1$ KEFs are zero on the separatrix (green contours).

## 3.2 Challenges and Solutions

While the examples above show cases where simple optimization leads to the separatrix, there are several crucial implementation details of our proposed methods. In particular, even a $\psi(x)$ that satisfies the Koopman PDE may fail to identify the true separatrix. This arises from known degeneracies in Koopman eigenfunctions, particularly in multistable or high-dimensional systems. To enable utilization of our tool, we describe two key failure modes and our strategies to resolve them, as summarized in Figure 4.

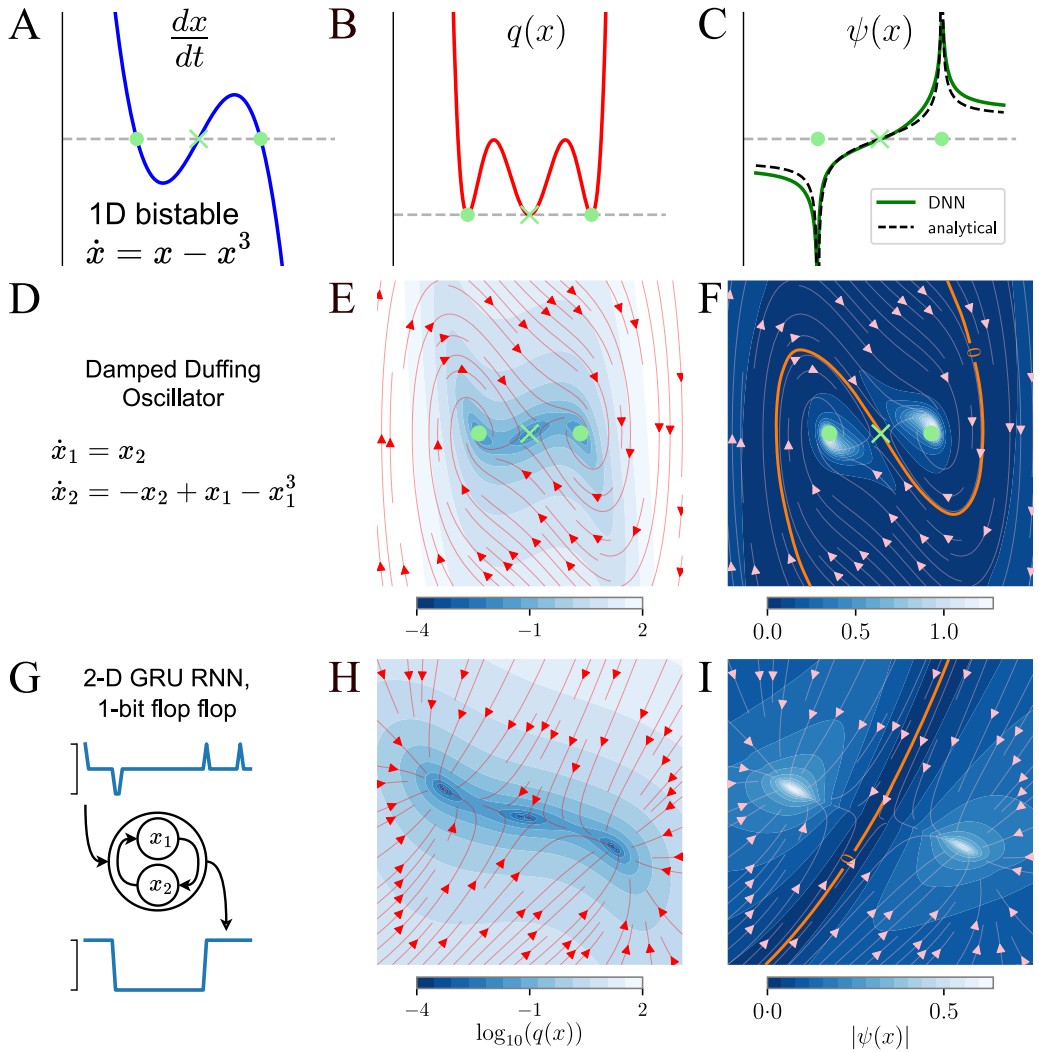

Figure 3: Our method to approximate KEFs in three bistable systems. (A) A 1D system $\dot{x} = x - x^3$. The curve shows $f(x)$, and its fixed points in light green – 'o's stable and 'x's unstable. (B) The kinetic energy $q(x)$ of this system. (C) the true KEF (8) and its DNN approximation obtained by our method. (D,E,F) Damped Duffing Oscillator in 2D (G,H,I) 2-unit GRU [37] RNN trained on 1-bit flip flop (1BFF) [5] and our KEFs.

**Degeneracy across basins.** A central issue stems from the compositional properties of Koopman eigenfunctions. Let $\psi_1(x)$ and $\psi_2(x)$ be eigenfunctions with eigenvalues $\lambda_1$ and $\lambda_2$. Then, their product is also a KEF:

$$\nabla[\psi_1(x)\psi_2(x)] \cdot f(x) = (\lambda_1 + \lambda_2)\psi_1(x)\psi_2(x). \tag{9}$$

In particular, consider a smooth KEF $\psi^1$ with $\lambda = 1$ that vanishes only on the separatrix (e.g., as in Figure 3). Now, consider a piecewise-constant function $\psi^0$ with $\lambda = 0$ that takes constant values within each basin and may be discontinuous at the separatrix. The product $\psi^1\psi^0$ remains a valid KEF with $\lambda = 1$, but it can now be zero across entire basins—thereby destroying the separatrix structure we aim to capture (Figure 4 top).

We observe this behavior empirically in Appendix D, where independently initialized networks converge to different spurious solutions. To mitigate this, we introduce a *balance regularization* term that biases $\psi$ to have nonzero values in opposing basins, encouraging sign changes across the

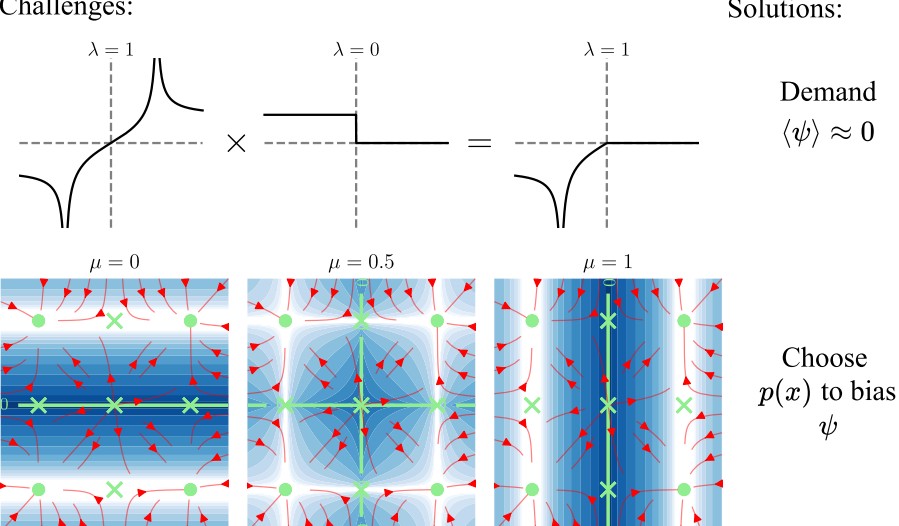

Figure 4: Top: In the presence of multiple basins, a KEF can collapse to zero within a single basin. This degeneracy is realised by multiplying the KEF with a piecewise constant KEF with $\lambda = 0$ and invoking (9). This example corresponds to $\dot{x} = x - x^3$. We introduce a regularisation term (10) to encourage the mean value $\langle \psi \rangle \approx 0$. This encourages solutions with sign changes across basins. Bottom: In higher dimensions, degeneracy arises from directional ambiguity in solutions. We visualise the analytical solution (12) for $\dot{x} = x - x^3; \dot{y} = y - y^3$. We address this by sampling from multiple local distributions around separatrix points and training an ensemble of KEFs.

separatrix. Specifically, we define:

$$\mathcal{L}_{\text{bal}} = \frac{(\mathbb{E}[\psi(x)])^2}{\text{Var}[\psi(x)]}, \tag{10}$$

and train using the combined loss $\mathcal{L}_{\text{ratio}} + \gamma_{\text{bal}}\mathcal{L}_{\text{bal}}$, where $\gamma_{\text{bal}}$ is a scalar hyperparameter.

In higher-dimensional systems, the Koopman PDE admits a family of valid KEFs that differ in their directional dependence. Consider a separable 2D system:

$$\dot{x} = f_1(x), \quad \dot{y} = f_2(y). \tag{11}$$

Solving the PDE for this system (appendix B) yields a family of KEFs parameterised by $\mu \in \mathbb{R}$:

$$\psi(x, y) = A(x)^\mu B(y)^{1-\mu}, \tag{12}$$

where $A(x)$ and $B(y)$ are KEFs to the respective 1D problems. For example, when $\mu = 1$, the eigenfunction depends only on $x$ and ignores $y$ – therefore unable to capture $y$-dependent separatrices. Figure 4 (bottom) illustrates this effect: different values of $\mu$ yield KEFs aligned with different separatrices.

Even in non-separable systems, this degeneracy can arise. Optimizing $\mathcal{L}_{\text{total}}$ can lead to a KEF that identifies some separatrices and ignores others (Appendix D). To address this, we train multiple KEFs $\{\psi_i(x)\}_{i=1}^k$, while using their input distributions to bias each one to capture a different separatrix. For $\psi_i(x)$, we choose two points in different basins of attraction, and then use a binary search on the line connecting them to find a point on the separatrix. Note that the KEF is still needed to obtain the full separatrix, and not just a point. Around each such point $\beta_i$, we define a local distribution $\mathcal{N}(\beta_i, \sigma_{ij}^2 I)$, using a range of scales $\{\sigma_{ij}\}_{j=1}^J$ to span both fine and global structure. For each distribution, we minimize the sum $\sum_{j=1}^J \mathcal{L}_{\text{total}}^j$. We then consider the union of the separatrices obtained from each of the KEFs to complete the picture (see 2-bit flip flop demonstration below).

### 3.3 Demonstrations

We demonstrate the applicability of the method on several qualitative examples.

A        B        C

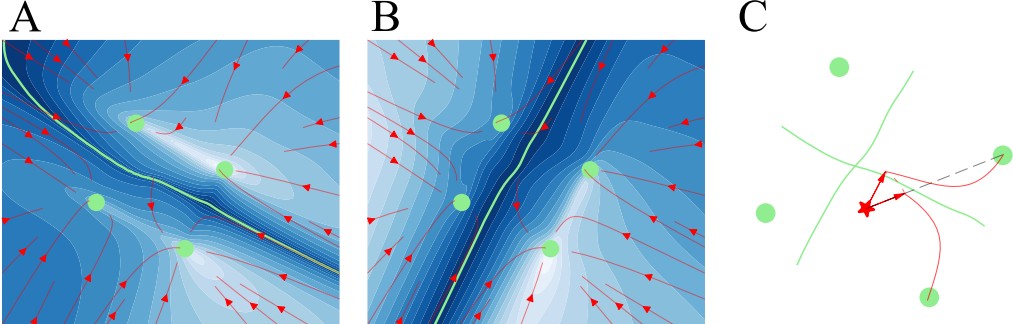

Figure 5: Two-bit flip flop task in a 3-unit GRU. The system has 4 stable fixed points (light-green points). (A,B) Two KEFs obtained by our method. They complement each other as they each captures a separatrix along one direction. (C) Use of KEF to design minimal perturbations that push trajectories across the separatrix.

### 3D GRU RNN Performing Two-Bit Flip Flop

We first demonstrate our method on a low-dimensional recurrent neural network trained to perform a two-bit flip flop (2BFF) task. Specifically, we use a 3-unit gated recurrent unit (GRU) network [37]. The trained network exhibits four stable fixed points (Figure 5), corresponding to different memory states of the task.

To overcome the degeneracies described in Figure 4, we adopt a targeted sampling strategy. We first identify points on the separatrix by interpolating between pairs of fixed points and performing binary search: at each step, we simulate the dynamics to determine basin membership and refine the search. Around these discovered separatrix points, we construct concentric isotropic Gaussian distributions, and sample from them to train on the loss $\mathcal{L}_{\text{total}}$ (Appendix G).

Two resulting KEF are shown in Figure 5 A,B). As expected, the KEFs vanish precisely along the separatrices. This result validates the ability of our method to recover boundary manifolds in neural dynamical systems, even in the presence of degeneracy. Once we know the separatrices, we can determine optimal perturbation directions (Figure 5C). Starting from a given initial condition (red star), we see that the same amplitude perturbation is sufficient to reach a different attractor when using the separatrix information, and insufficient when directed at the desired attractor. A more quantitative depiction of this effect is shown below in a higher-dimensional system.

### 11D Ecological Dynamics

We next apply our method (Appendix G) to a high-dimensional ecological model: a generalized Lotka–Volterra (gLV) system fit to genus-level abundance data from a mouse model of antibiotic-induced *Clostridioides difficile* infection (CDI) [38]. The system has five stable fixed points. Following [39] we focus our analysis to two of these fixed points representing healthy and diseased microbial states.

We optimize the KEF in the full 11-dimensional state space. For interpretability, we follow the projection approach of [39], visualizing the dynamics in the 2D plane spanned by the two chosen stable fixed points and the origin (see Figure 6). Although the KEF is trained entirely in the original 11-dimensional space, its zero level set (light green curve) aligns well with the true separatrix (orange line) computed using a grid of initial conditions in the 2D slice [39].

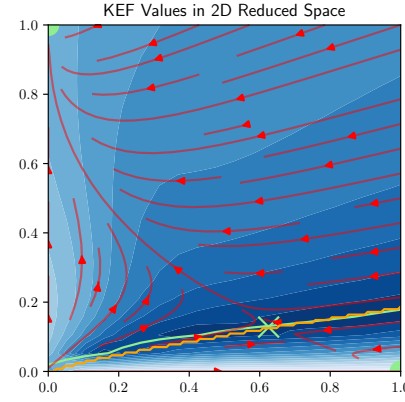

KEF Values in 2D Reduced Space

Figure 6: KEF approximation in a fitted 11D gLV model of CDI [38, 39]. Zero level set of the KEF aligns with the separatrix in a 2D projection plane.

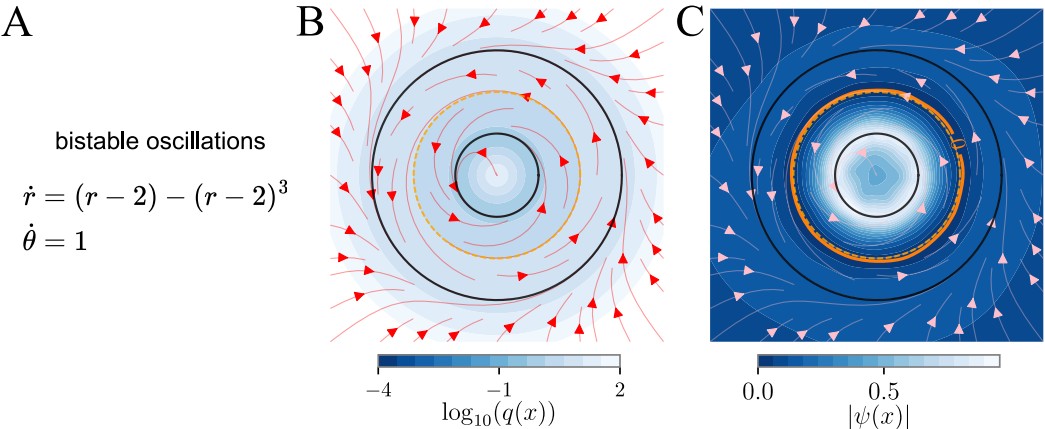

Figure 7: Applying our method to a system of stable and unstable limit cycles, a system without any fixed points on the separatrix. (A) system equations. (B) kinetic energy, with dashed line for separatrix. (C) KEF from our method with zero level highlighted.

This result demonstrates that our technique can be applied directly to real-world fitted models, without dimensionality reduction at training time.

**Limit cycle separatrix**

We test our method in a setting where there are no fixed points along the separatrix. We construct a system which oscillates at a fixed frequency ($\dot{\theta} = 1$), but converges to one of two preferred amplitudes ($\dot{r} = (r-2) - (r-2)^3$). The system has three limit-cycles, two of them stable ($r = 1, 3$) and one unstable ($r = 2$). In Figure 7B we visualise the flow, its kinetic energy and the limit cycles. The system has no fixed points, and thus fixed point analysis is futile. We utilize Radial basis function neural network [40] to parameterise the KEF (Appendix F).

We show that our approximation of the KEF recovers the separatrix at $r = 2$ (Figure 7C).

**668D RNN fit to mouse neural activity**

To demonstrate our method in a high-dimensional (see Appendix H for scaling results) and neuroscientifically relevant setting, we applied it to a recurrent neural network (RNN) trained to reproduce mouse neural activity from Finkelstein et al. [9]. The trained RNN exhibits bistability between two memory states. As in lower-dimensional systems, we first located a point on the separatrix by performing a binary search along the line connecting the fixed-point attractors, simulating the dynamics at each step to determine basin membership.

In the original experiment, mice were trained to respond to optogenetic stimulation of their sensory cortices, and the RNN was fit to the peristimulus time histogram of recorded neural activity. The network undergoes a bifurcation from monostability to bistability as a function of an external ramping input $u_{\text{ext}}$. For analysis, we fixed $u_{\text{ext}} = 0.9$ within the bistable regime and trained the Koopman eigenfunction (KEF) network (Appendix G) using samples drawn from isotropic Gaussian distributions centered at the separatrix point.

Because of the high-dimensionality, direct visualization of the learned KEF and dynamics is not feasible. Instead, we validated the model using a curve-based evaluation approach. We construct multiple Hermite polynomial curves that interpolate between the two stable fixed points. The curvature of each curve is parameterized by a random vector, and each is defined by a parameter $\alpha \in [0, 1]$, where $\alpha = 0$ corresponds to one attractor and $\alpha = 1$ to the other (see appendix E). Because each curve continuously connects the fixed points, it must cross the separatrix. Figure 8A shows a 2D PCA projection of several such Hermite curves. Crucially, the actual curves span the entire 668D space. We simulate dynamics from 100 points along each curve and determine their final basin to infer where each curve crosses the separatrix, forming a ground-truth reference.

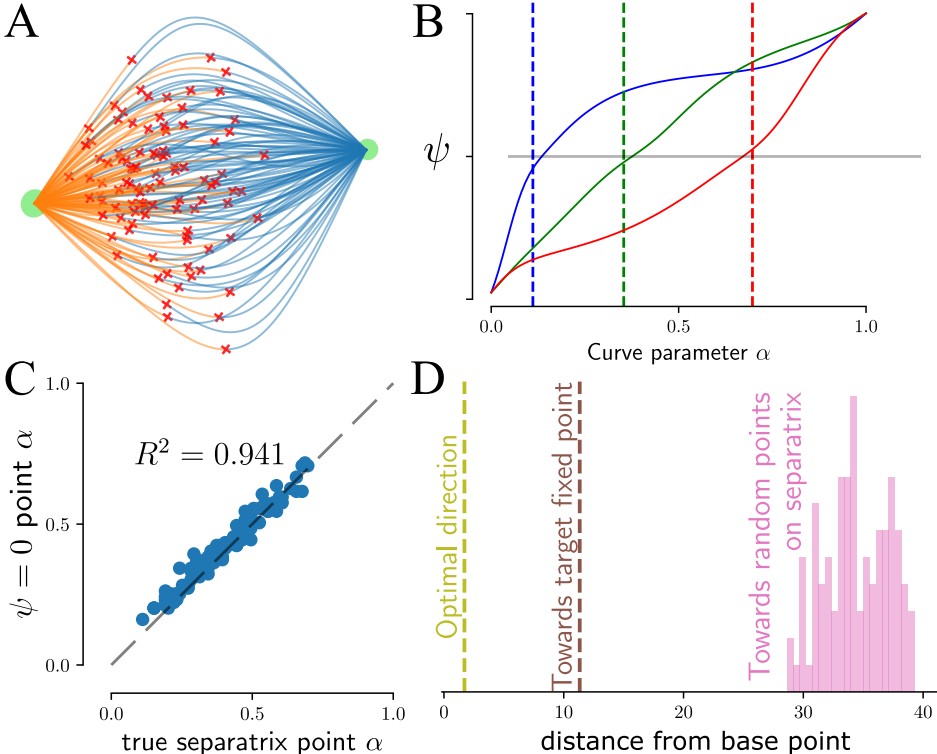

Figure 8: Validation of KEF approximation in a 668D RNN fit to mouse neural activity [9]. (A) PCA projection of Hermite curves between fixed points coloured by true basin labels and separatrix points along the curves (red crosses). (B) KEF values along three Hermite curves versus curve parameter $\alpha$, as well the true separatrix point along the curve. (C) Comparison between true and predicted separatrix positions along curves. (D) Perturbation amplitudes $\|\boldsymbol{\Delta}\|$, i.e., distance from $x_{\text{base}}$ to perturbation targets. The KEF-guided solution yields the smallest perturbation crossing the separatrix.

Next, we evaluate the learned KEF along these same curves. Figure 8B shows KEF values along sample Hermite curves as a function of $\alpha$, with the zero crossing indicating our predicted separatrix. Figure 8C compares the $\alpha$-locations of the ground truth and the KEF-predicted separatrix points. We observe strong agreement, indicating that the learned KEF reliably tracks the separatrix in this high-dimensional system.

Finally, we demonstrate how the KEF can be used to design minimal perturbations that shift the state across the separatrix (similar to Figure 1A, Figure 5C). In general, this involves an input-driven dynamics $\dot{\boldsymbol{x}} = \tilde{f}(\boldsymbol{x}, \boldsymbol{u})$, with time-varying inputs $\boldsymbol{u}(t)$. To demonstrate the utility of the method we study a specific, simplified scenario in which the input is a strong instantaneous perturbation $\boldsymbol{u}(t) := \boldsymbol{\Delta}\delta(t)$, which moves the state $\boldsymbol{x}(0_+) = \boldsymbol{x}(0_-) + \boldsymbol{\Delta}$, after which the dynamics evolves according to $f(\boldsymbol{x}) := \tilde{f}(\boldsymbol{x}, 0)$.

Given a base point $\boldsymbol{x}(0_-) := \boldsymbol{x}_{\text{base}}$, just before perturbation, we aim to solve:

$$\boldsymbol{\Delta}^* = \arg\min_{\boldsymbol{\Delta}} \|\boldsymbol{\Delta}\|_2^2 \quad \text{subject to} \quad |\psi(\boldsymbol{x}_{\text{base}} + \boldsymbol{\Delta})| = 0. \tag{13}$$

Taking advantage of the differentiable nature of $\psi$ in our method, we use the Adam optimizer [41] to find $\boldsymbol{\Delta}^*$ from random initialization. Figure 8D shows that indeed the optimised perturbation $\boldsymbol{\Delta}^*$ is smaller, compared to simply aiming the perturbation towards the target fixed point, or towards random points on the separatrix.

# 4 Discussion

We presented a novel framework for identifying separatrices in high-dimensional, black-box dynamical systems using Koopman eigenfunctions (KEFs). This method is particularly useful for analyzing recurrent neural networks (RNNs), which are commonly used to model neural computations involving multiple stable states.

Prior efforts in reverse-engineering RNNs relied heavily on locating fixed points and linearizing dynamics locally [6–16]. While powerful, these methods cannot directly capture global structures or predict system responses to large perturbations that cross basin boundaries. By directly approximating scalar-valued KEFs that vanish precisely on separatrices, our method complements and extends existing local linearization approaches. Practitioners can use our KEFs alongside fixed-point analysis to achieve a comprehensive understanding of the dynamical system's landscape.

While FTLE methods [31–33] also identify separatrices as ridges of finite-time trajectory divergence, they may change sharply near separatrices while providing little gradient elsewhere. We speculate that this could limit their usefulness in high-dimensional systems, where gradient-based localization is needed. Moreover, differentiating FTLE requires differentiating through the dynamical function, which may be computationally expensive or even infeasible due to vanishing/exploding gradients. In contrast, our Koopman eigenfunction framework, by integrating globally, provides a smooth scalar field whose zero level set identifies the separatrix, enabling efficient gradient-based searches without repeated forward simulations of the target system.

Our work also advances the application of Koopman operator theory to dynamical systems. Previous studies primarily utilized Koopman eigenfunctions to predict or control dynamics within a single basin of attraction [28, 42–45]. Likewise, methods comparing dynamical systems to one another use the dynamic mode decomposition which does not always discern between different basins [19, 20, 46]. Such studies usually involve KEFs associated with negative eigenvalues ($\lambda < 0$), which exhibit opposite behavior to ours: they explode at separatrices and approach zero at attractors. In contrast, we specifically targeted eigenfunctions associated with positive eigenvalues, ensuring their zeros correspond exactly to separatrices.

To help practitioners use our method, we highlight inherent challenges, such as degeneracy in the Koopman PDE. To overcome these, we introduced a specific regularization–a balance term ensuring eigenfunctions change sign across different basins and show how to choose distributions for the Koopman PDE. These ideas build on existing work on KEF approximation [30, 29, 47], enabling reliable identification of separatrices in diverse and high-dimensional systems.

Our method provides an alternative to more direct approaches for locating separatrices, such as grid searches or bisection methods that repeatedly simulate the ODE from many initial conditions [39]. While learning a KEF involves iteratively solving a PDE over phase space, trajectory-based approaches scale with simulation time and often revisit the same regions of phase space. In contrast, solving the PDE resembles dynamic programming and can be made more efficient (Appendix H), particularly near separatrices where critical slowing down makes ODE-based methods computationally demanding.

While we demonstrated the applicability of our method to diverse scenarios, we do not provide theoretical guarantees linking the accuracy of the KEF approximation and that of the separatrix location. Furthermore, like techniques for finding fixed points [5, 48], our method requires knowing the dynamics in the entire phase space. Extending this to trajectory-based methods [49, 50, 19] can facilitate separatrix inference directly from neural data.

An interesting extension of our work is to stochastic dynamical systems [51, 52]. While one can approximate separatrices using only the deterministic component of the dynamics, a full treatment of stochasticity and its effect on basin boundaries remains open. Likewise, data-driven models with uncertainty in the inferred flow [53] raise questions about how this uncertainty propagates to the separatrix. Since separatrices often occupy sparsely sampled or unstable regions of phase space, this motivates active sampling—potentially through targeted optogenetic stimulations—to improve model accuracy along these boundaries [54].

In conclusion, we hope that by focusing on separatrices, our method could inform intervention strategies in neuroscience, ecological or engineering systems, providing a general-purpose tool to predict and control transitions between stable states in complex dynamical landscapes.

## Acknowledgements

This work was supported by the Israel Science Foundation (grant No. 1442/21 to OB) and Human Frontiers Science Program (HFSP) research grant (RGP0017/2021 to OB). The funders had no role in study design, data collection and analysis, decision to publish, or preparation of the manuscript.

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

# A Analytical KEF derivation in 1D bistable system

We would like to find an analytical Koopman eigenfunction for the scalar dynamical system:

$$\dot{x} = x - x^3 \tag{14}$$

In the 1D case, the Koopman PDE (3) reduces to a first-order ordinary differential equation

$$\frac{d\psi}{dx} f(x) = \lambda \psi(x). \tag{15}$$

With $f(x) = x - x^3$ and $\lambda = 1$ we have:

$$\psi'(x)(x - x^3) = \psi(x) \tag{16}$$

$$\Rightarrow \quad \frac{\psi'(x)}{\psi(x)} = \frac{1}{x - x^3} \tag{17}$$

To solve this integral we first simplify the integrand.

$$x - x^3 = x(1 - x^2) = x(1 - x)(1 + x) \tag{18}$$

So,

$$\frac{1}{x(1 - x)(1 + x)} = \frac{A}{x} + \frac{B}{1 - x} + \frac{C}{1 + x} \tag{19}$$

Solving for $A$, $B$, $C$ yields $A = 1$, $B = \frac{1}{2}$, $C = -\frac{1}{2}$.

Now we can integrate,

$$\int \frac{1}{x - x^3} \, dx = \int \left( \frac{1}{x} + \frac{1}{2(1 - x)} - \frac{1}{2(1 + x)} \right) dx \tag{20}$$

$$= \log |x| - \frac{1}{2} \log |1 - x| - \frac{1}{2} \log |1 + x| + C \tag{21}$$

$$\log \psi(x) = \log |x| - \frac{1}{2} \log |1 - x| - \frac{1}{2} \log |1 + x| + C \tag{22}$$

$$\Rightarrow \quad \psi(x) = C' \cdot \frac{|x|}{\sqrt{|1 - x^2|}} \tag{23}$$

To bring it into the form in the main text we use the product composition rule (9). We can multiply our solution by the $\text{sign}(x)$ function which is a $\lambda = 0$ eigenfunction because it remains constant in each basin (see Figure 4A). In other words, we flip the sign of our solution $\psi(x) \to -\psi(x)$ for $x < 0$.

$$\psi(x) = C' \frac{x}{\sqrt{|1 - x^2|}} \tag{24}$$

and this remains a KEF with $\lambda = 1$.

# B Eigenfunction Degeneracy in higher dimensions

Consider a separable 2D dynamical system:

$$\dot{x} = f_x(x), \tag{25}$$
$$\dot{y} = f_y(y), \tag{26}$$

which we write compactly as:

$$\dot{\mathbf{x}} = \mathbf{f}(x, y) = \begin{bmatrix} f_x(x) \\ f_y(y) \end{bmatrix}. \tag{27}$$

We seek a Koopman eigenfunction $\psi(x, y)$ satisfying:

$$\nabla\psi \cdot \mathbf{f}(x, y) = \lambda\psi(x, y). \tag{28}$$

Assume $\lambda = 1$ and a separable form $\psi(x, y) = X(x)Y(y)$. Then:

$$\frac{\partial\psi}{\partial x} = X'(x)Y(y), \tag{29}$$

$$\frac{\partial\psi}{\partial y} = X(x)Y'(y), \tag{30}$$

$$\nabla\psi \cdot \mathbf{f} = X'(x)Y(y)f_x(x) + X(x)Y'(y)f_y(y) = X(x)Y(y). \tag{31}$$

Dividing both sides by $X(x)Y(y)$ gives:

$$\frac{X'(x)}{X(x)}f_x(x) + \frac{Y'(y)}{Y(y)}f_y(y) = 1. \tag{32}$$

The above equation requires that the sum of the above two terms, which each depend on different variables must be 1 for all $x$, $y$. It follows that each term is also a constant function.

$$\frac{X'(x)}{X(x)}f_x(x) = \mu, \tag{33}$$

$$\frac{Y'(y)}{Y(y)}f_y(y) = 1 - \mu, \tag{34}$$

for an arbitrary constant $\mu \in \mathbb{R}$.

Define the antiderivatives:

$$A(x) = \int \frac{1}{f_x(x)}dx, \tag{35}$$

$$B(y) = \int \frac{1}{f_y(y)}dy. \tag{36}$$

Then the logarithms of the separated components are:

$$\log X(x) = \mu A(x) \quad \Rightarrow \quad X(x) = \left(e^{A(x)}\right)^\mu, \tag{37}$$

$$\log Y(y) = (1 - \mu)B(y) \quad \Rightarrow \quad Y(y) = \left(e^{B(y)}\right)^{1-\mu}. \tag{38}$$

Thus, the general separable Koopman eigenfunction is:

$$\psi(x, y) = \left(e^{A(x)}\right)^\mu \cdot \left(e^{B(y)}\right)^{1-\mu}. \tag{39}$$

## C    Relation of our definition to the Koopman Operator

In the main text, we introduced Koopman eigenfunctions as scalar functions $\psi : \mathcal{X} \to \mathbb{R}$ that evolve exponentially along trajectories $\boldsymbol{x}(t) \in \mathcal{X}$ of a dynamical system $\dot{\boldsymbol{x}} = f(\boldsymbol{x})$:

$$\frac{d}{dt}\psi(\boldsymbol{x}(t)) = \lambda\psi(\boldsymbol{x}(t)). \tag{40}$$

Here, we clarify the origin of this equation by defining the Koopman operator, linking our approach to the broader theory.

Let $g : \mathcal{X} \to \mathbb{R}$ be a real-valued function of the system state—commonly referred to as an *observable*. The collection of such observables forms an infinite-dimensional function space, typically a Hilbert space once equipped with an inner product $\langle g, g' \rangle$. The Koopman operator acts linearly on this space.

*Remark.* When studying nonlinear dynamical systems with multiple basins of attraction, as we do, the corresponding Koopman eigenfunctions are generally *not* square-integrable, and therefore fall outside the Hilbert space defined by the standard inner product. We are aware of this theoretical limitation and continue to employ the Koopman framework regardless. In practice, neural networks learn finite, smooth approximations to these otherwise singular structures.

For a continuous-time system, the Koopman operator $\mathcal{K}_\tau$ evolves observables according to the flow map $F_\tau : \mathcal{X} \to \mathcal{X}$, which advances the state forward by time $\tau$:

$$(\mathcal{K}_\tau g)\big(\boldsymbol{x}(t)\big) = g\Big(F_\tau\big(\boldsymbol{x}(t)\big)\Big) = g\big(\boldsymbol{x}(t + \tau)\big). \tag{41}$$

The infinitesimal generator of the Koopman semigroup $\{\mathcal{K}_\tau\}_{\tau \geq 0}$, often denoted simply as $\mathcal{K}$, is defined as:

$$\mathcal{K}g := \lim_{\tau \to 0} \frac{\mathcal{K}_\tau g - g}{\tau} = \lim_{\tau \to 0} \frac{g\Big(F_\tau(\boldsymbol{x})\Big) - g(\boldsymbol{x})}{\tau}. \tag{42}$$

When evaluated along a trajectory $\boldsymbol{x}(t)$, this yields:

$$\mathcal{K}g\big(\boldsymbol{x}(t)\big) = \lim_{\tau \to 0} \frac{g\big(\boldsymbol{x}(t + \tau)\big) - g\big(\boldsymbol{x}(t)\big)}{\tau} \tag{43}$$

$$= \frac{d}{dt} g\big(\boldsymbol{x}(t)\big) = \nabla g\big(\boldsymbol{x}(t)\big) \cdot \dot{\boldsymbol{x}}(t) = \nabla g\big(\boldsymbol{x}(t)\big) \cdot f\Big(\boldsymbol{x}(t)\Big). \tag{44}$$

This operator is also known as the *Lie derivative* of $g$ along the vector field $f$.

Thus, an eigenfunction $\psi$ of $\mathcal{K}$ satisfying

$$\mathcal{K}\psi = \lambda\psi \tag{45}$$

recovers the Koopman eigenfunction equation (3) used in the main text.

## D  KEF degeneracy in randomly initialised DNN solutions

Main text Figure 4 illustrates challenges arising due to the degeneracy of the Koopman PDE (3). In Figure 9, we train several DNNs on a 2-unit GRU trained on the 2BFF. Each DNN is independently initialised and trained on a single distribution without the balance regularisation term $\mathcal{L}_{\text{bal}}$, i.e., $\gamma_{\text{bal}} = 0$. The resulting KEF approximations exhibit the same modes of degeneracy - zero on certain basins as well as vertical and horizontal variants.

## E  Curve-based validation approach

In high-dimension, we cannot visualize the entire phase space to check whether zeros of the KEF coincide with the separatrix. Instead, we generate a family of smooth curves that connect two attractors, and hence must pass through a separatrix. In the 64D GRU flip flop example, the two attractors are the two stable fixed points $x, y \in \mathbb{R}^N$. We use *cubic Hermite interpolation* with randomized tangent vectors at the endpoints. Each curve is defined by:

$$H(\alpha) = h_{00}(\alpha)\, x + h_{10}(\alpha)\, m_x + h_{01}(\alpha)\, y + h_{11}(\alpha)\, m_y, \quad \alpha \in [0, 1] \tag{46}$$

where the Hermite basis functions are:

$$h_{00}(\alpha) = 2\alpha^3 - 3\alpha^2 + 1, \tag{47}$$

$$h_{01}(\alpha) = -2\alpha^3 + 3\alpha^2, \tag{48}$$

$$h_{10}(\alpha) = \alpha^3 - 2\alpha^2 + \alpha, \tag{49}$$

$$h_{11}(\alpha) = \alpha^3 - \alpha^2. \tag{50}$$

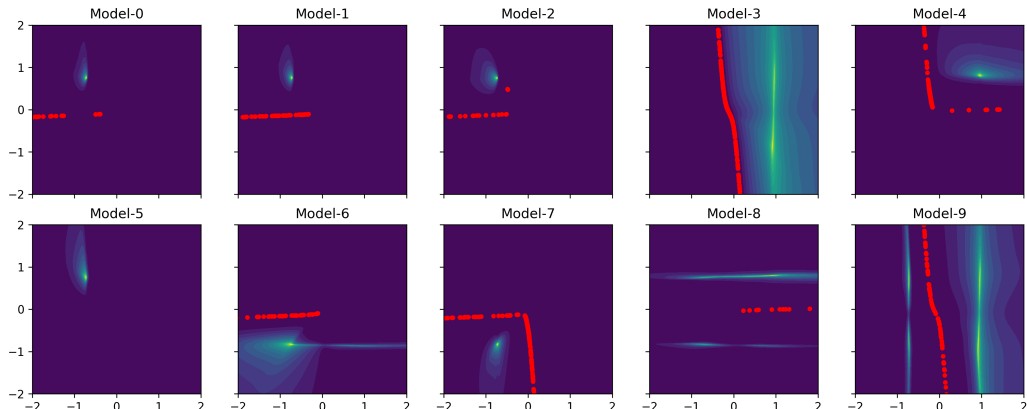

Figure 9: Many KEFs of to for 2 bit flip flop in 2D

Notice that $H(0) = x$ and $H(1) = y$.

The tangent vectors $m_x$ and $m_y$ are initialized as $y - x$ and perturbed with Gaussian noise:

$$m_x = (y - x) + \epsilon_x, \quad m_y = (y - x) + \epsilon_y, \quad \epsilon_x, \epsilon_y \sim \mathcal{N}(0, \sigma^2 I). \tag{51}$$

We sample multiple such curves with independently drawn perturbations. This produces a family of curves that interpolate between $x$ and $y$, while varying in geometry, enabling randomized exploration of intermediate regions in state space. Crucially, the curves are not limited to the manifold spanned by the attractors, but extend to all dimensions (controlled by $\sigma$. Optional constraints (e.g., non-negativity) can be imposed by rejecting any curve that violates them. For each such curve, we both evaluate the KEF and simulate the ODE to determine the position of the separatrix.

## F  Neural network architectures

In most of the demonstrations we use a ResNet architecture [55] with a $\tanh$ activation function.

Let the input to the network be $\mathbf{x}_{\text{in}} \in \mathbb{R}^{d_{\text{in}}}$, and let the hidden activations be $\mathbf{x}^{(\ell)} \in \mathbb{R}^{d_{\text{hid}}}$, with output $\mathbf{x}_{\text{out}} \in \mathbb{R}^{d_{\text{out}}}$ and $L$ layers.

The network receives inputs at the first layer as $\mathbf{x}^{(0)} = \text{Pad}(a\,\mathbf{x}_{\text{in}})$, where $\text{Pad}$ appends zeros to the input (we always choose $d_{\text{hid}} > d_{\text{in}}$), and $a \in \mathbb{R}_+$ is a scalar chosen so that the elements of $\mathbf{x}^{(0)}$ are $\mathcal{O}(1)$. The network then updates the hidden state at each layer $\ell$ as

$$\mathbf{x}^{(\ell+1)} = \mathbf{x}^{(\ell)} + \tanh\left(W^{(\ell)}\mathbf{x}^{(\ell)} + \mathbf{b}^{(\ell)}\right), \quad \ell = 0, \dots, L-1, \tag{52}$$

where $W^{(\ell)} \in \mathbb{R}^{d_{\text{hid}} \times d_{\text{hid}}}$ and $\mathbf{b}^{(\ell)} \in \mathbb{R}^{d_{\text{hid}}}$.

The output is obtained by applying a final linear layer:

$$\mathbf{x}_{\text{out}} = W^{\text{out}}\mathbf{x}^{(L)} + \mathbf{b}^{\text{out}}, \tag{53}$$

where $W^{\text{out}} \in \mathbb{R}^{d_{\text{out}} \times d_{\text{hid}}}$ and $\mathbf{b}^{\text{out}} \in \mathbb{R}^{d_{\text{out}}}$. During optimization, gradients $\nabla_\theta \mathcal{L}_{\text{total}}$ are computed for all parameters $\theta = (W^{(0:L-1)}, \mathbf{b}^{(0:L-1)}, W^{\text{out}}, \mathbf{b}^{\text{out}})$.

**Choices for each system**

For the results in Figures 3, 5, 8, we use $L = 20$, $d_{\text{hid}} = 400$. $d_{\text{out}} = 1$ and $d_{\text{in}} = N$ the dimension of the dynamical system. For the gLV system in Figure 6 we use $L = 25$ and $d_{\text{hid}} = 1000$.

**Radial Basis Function (RBF) Layer**

For the limit cycles example in Figure 7 we use a single Radial Basis Function layer [40].

Given an input $\mathbf{x} \in \mathbb{R}^{d_{\text{in}}}$, the RBF layer maps it to an output $\mathbf{y} \in \mathbb{R}^{d_{\text{out}}}$ through a set of $M$ radial basis functions, each centered at $\mathbf{c}_i \in \mathbb{R}^{d_{\text{in}}}$, with a shape parameter $\varepsilon_i > 0$ and linear combination weights $a_{ij} \in \mathbb{R}$.

To compute RBF activations for $i = 1, \ldots, M$, we define the scaled radial distance:

$$s_i(\mathbf{x}) = \varepsilon_i \cdot \|\mathbf{x} - \mathbf{c}_i\|, \tag{54}$$

and then apply a gaussian radial basis function

$$\varphi_i(\mathbf{x}) = \exp(-s_i(\mathbf{x})^2). \tag{55}$$

The final output is a linear combination of the basis activations:

$$y_j(\mathbf{x}) = \sum_{i=1}^{M} a_{ji} \cdot \varphi_i(\mathbf{x}), \quad j = 1, \ldots, d_{\text{out}} \tag{56}$$

We use $M = 300$, and $d_{\text{out}} = 1$. During optimization, gradients $\nabla\theta\mathcal{L}_{\text{total}}$ are computed for all parameters $\theta = (\{a_{ji}\}, \{\mathbf{c}_i\}, \{\varepsilon_i\})$.

## G  Optimisation

Our optimisation procedure described in the main text is summarised in Algorithm 1. It implements the training of the neural network Koopman eigenfunction using the ratio loss and balance regularisation, for multiple sampling distributions.

We minimise the total loss:

$$\mathcal{L}_{\text{total}} = \sum_{j=1}^{J} \mathcal{L}_{\text{ratio}}^j + \gamma_{\text{bal}}\mathcal{L}_{\text{bal}}^j. \tag{57}$$

where $j$ corresponds to the $j^{\text{th}}$ sampling distribution (see main text section 3.2). $B$ $N$-dimensional points in the state space $\mathcal{X}$ are sampled from each distribution $\boldsymbol{x}_i^j \sim p_j(\boldsymbol{x})$. The ratio loss is the Koopman PDE error, normalised by a sample-shuffled version:

$$\mathcal{L}_{\text{ratio}}^j = \frac{\sum_{i=1}^{B}(\text{LHS}_i^j - \text{RHS}_i^j)^2}{\sum_{i=1}^{B}(\text{LHS}_i^j - \text{RHS}_{\text{perm}(i)}^j)^2} \tag{58}$$

$$\text{LHS}_i^j = \nabla\psi(\boldsymbol{x}_i^j) \cdot f(\boldsymbol{x}_i^j) \qquad \text{left-hand-side of the Koopman PDE (3)} \tag{59}$$

$$\text{RHS}_i^j = \lambda\psi(\boldsymbol{x}_i^j) \qquad \text{right-hand-side of the Koopman PDE (3)} \tag{60}$$

where $\text{perm}(i)$ is a random permutation of the numbers $1, 2, \ldots, B$ sampled during each training iteration.

The balance regularisation loss is the squared mean of the KEF values divided by their variance:

$$\mathcal{L}_{\text{bal}}^j = \frac{(\bar{\psi}^j)^2}{\frac{1}{B}\sum_{i=1}^{B}(\psi(\boldsymbol{x}_i^j) - \bar{\psi}_j)^2}, \tag{61}$$

$$\bar{\psi}^j = \frac{1}{B}\sum_{i=1}^{B}\psi(\boldsymbol{x}_i^j). \tag{62}$$

In general we set $\gamma_{\text{bal}} = 0.05$. For the limit cycles Figure 7 we set $\gamma_{\text{bal}} = 0$.

We compute $\nabla\psi(x)$ using Pytorch's `torch.autograd.grad`, specifying `create_graph=True`, since we differentiate through this a second time to compute the gradients $\nabla_\theta\mathcal{L}_{\text{total}}$ with respect to the neural network parameters $\theta$.

We use the Adam optimiser [41] with learning rate $10^{-4}$ and l2 normalisation $10^{-5}$. We use $B = 1000$ and train for 1000 iterations.

Only in the case of the 11D gLV, Figure 6 we use $B = 5000$ and train for 5000 iterations.

A summary of all hyperparameters is provided in Table 1.

---

**Algorithm 1** Train Koopman Eigenfunction Network

---

**Require:** Sampling distributions $\{p_j(\boldsymbol{x})\}_{j=1}^J$; vector field $f$; eigenvalue $\lambda$; neural network architecture $\psi_\theta$; batch size $B$; iterations $T$; balance weight $\gamma_{\text{bal}}$; learning rate $\eta$; small $\varepsilon = 10^{-12}$

1: Initialize $\theta$
2: **for** $t = 1 \rightarrow T$ **do**
3:      $L_{\text{total}} \leftarrow 0$
4:      **for** $j = 1 \rightarrow J$ **do**
5:          $\{\boldsymbol{x}_i^j, \psi_i^j, \nabla\psi_i^j\} \leftarrow \textsc{SampleAndEvaluate}(p_j, \psi_\theta, B)$
6:          $L_{\text{ratio}}^j \leftarrow \textsc{ComputeRatioLoss}(\{\psi_i^j, \nabla\psi_i^j, f(\boldsymbol{x}_i^j), \lambda\})$
7:          $L_{\text{bal}}^j \leftarrow \textsc{ComputeBalanceLoss}(\{\psi_i^j\})$
8:          $L_{\text{total}} \leftarrow L_{\text{total}} + L_{\text{ratio}}^j + \gamma_{\text{bal}} L_{\text{bal}}^j$
9:      **end for**
10:      Compute gradients of $L_{\text{total}}$ w.r.t. $\theta$
11:      Update weights $\theta$ using gradients and learning rate $\eta$
12: **end for**
13: **return** trained parameters $\theta$
14: **procedure** $\textsc{SampleAndEvaluate}(p_j, \psi_\theta, B)$
15:      Sample $\{\boldsymbol{x}_i^j\}_{i=1}^B \sim p_j(\boldsymbol{x})$
16:      Compute $\psi_i^j \leftarrow \psi_\theta(\boldsymbol{x}_i^j)$
17:      Compute $\nabla\psi_i^j \leftarrow \nabla_{\boldsymbol{x}}\psi_\theta(\boldsymbol{x}_i^j)$
18:      **return** $\{\boldsymbol{x}_i^j, \psi_i^j, \nabla\psi_i^j\}$
19: **end procedure**
20: **procedure** $\textsc{ComputeRatioLoss}(\{\psi_i^j, \nabla\psi_i^j, f(\boldsymbol{x}_i^j), \lambda\})$
21:      Compute $\text{LHS}_i^j = \nabla\psi_i^j \cdot f(\boldsymbol{x}_i^j)$
22:      Compute $\text{RHS}_i^j = \lambda\,\psi_i^j$
23:      Draw random permutation $\pi$ of $\{1, \ldots, B\}$
24:      $n_j \leftarrow \sum_i (\text{LHS}_i^j - \text{RHS}_i^j)^2$
25:      $d_j \leftarrow \sum_i (\text{LHS}_i^j - \text{RHS}_{\pi(i)}^j)^2$
26:      **return** $L_{\text{ratio}}^j = n_j/(d_j + \varepsilon)$
27: **end procedure**
28: **procedure** $\textsc{ComputeBalanceLoss}(\{\psi_i^j\})$
29:      $\bar{\psi}^j = \frac{1}{B}\sum_i \psi_i^j$
30:      $v^j = \frac{1}{B}\sum_i (\psi_i^j - \bar{\psi}^j)^2$
31:      **return** $L_{\text{bal}}^j = (\bar{\psi}^j)^2/(v^j + \varepsilon)$
32: **end procedure**

---

Table 1: Algorithm details and hyperparameters for various systems. System dimensionality $N$, Koopman eigenvalue $\lambda$, balance regularisation weight $\gamma_{\text{bal}}$, batch-size $B$, training iterations $T$, learning rate $\eta$, ResNet depth $L$ and width $d_{\text{hid}}$, number of Radial Basis Functions $M$.

| Dynamical System | $N$ | $\lambda$ | $\gamma_{\text{bal}}$ | $B$ | $T$ | $\eta$ | $L$ | $d_{\text{hid}}$ | $M$ |
|---|---|---|---|---|---|---|---|---|---|
| Bistable 1D | 1 | 1 | 0.05 | 1000 | 1000 | $10^{-4}$ | 20 | 400 | – |
| Damped Duffing oscillator | 2 | 1 | 0.05 | 1000 | 1000 | $10^{-4}$ | 20 | 400 | – |
| 1BFF, 2D GRU | 2 | 1 | 0.05 | 1000 | 1000 | $10^{-4}$ | 20 | 400 | – |
| 2BFF, 3D GRU | 3 | 0.2 | 0.05 | 1000 | 1000 | $10^{-4}$ | 20 | 400 | – |
| 1BFF, 64D | 64 | 0.1 | 0.05 | 1000 | 1000 | $10^{-4}$ | 20 | 400 | – |
| Two Limit Cycles | 2 | 1 | 0 | 1000 | 1000 | $10^{-4}$ | – | – | 300 |
| Ecology gLV | 11 | 0.1 | 0.05 | 5000 | 5000 | $10^{-4}$ | 25 | 1000 | – |
| Data-trained RNN [9] | 668 | 0.02 | 0.05 | 1000 | 1000 | $10^{-4}$ | 7 | 1200 | – |

### G.1 Choice of Training Distributions

Choosing suitable training distributions $p_j(\boldsymbol{x})$ is an important step in applying our method. The distributions are selected to satisfy the following criteria:

1. They are approximately bisected by the separatrix, ensuring roughly equal sampling from both basins and enabling the balance loss to be satisfied.

2. They sample sufficiently near the attractors to capture the global bistable dynamics.

3. They are approximately forward-invariant, i.e., trajectories initialized from the distribution remain within its support when evolved forward in time. This avoids loss of mass due to transient amplification along unstable directions.

4. In systems with multiple spurious attractors, they avoid sampling from basins outside the domain of interest.

Below we list the specific distributions used for each system.

**1D bistable system (Fig. 3C):** $[\mathcal{N}(0, 1), \mathcal{N}(0, 3)]$.

**2D damped Duffing oscillator (Fig. 3F):** $\mathcal{N}(\mathbf{0}, \sigma_j^2 \mathbf{I}_2)$, with $\sigma_j = [0.1, 0.5, 1.0, 2.0]$, where $\mathbf{I}_2$ is the $2 \times 2$ identity matrix.

**2D 1-bit flip-flop GRU (Fig. 3I):** $\mathcal{N}(\mathbf{0}, \sigma_j^2 \mathbf{I}_2)$, with $\sigma_j = [0.01, 0.1, 0.5, 1.0, 2.0, 4.0]$.

**3D GRU 2-bit flip-flop (Fig. 5):** $\mathcal{N}(\boldsymbol{\mu}, \sigma_j^2 \mathbf{I}_3)$, with $\sigma_j = [0.01, 0.05, 0.2, 1.0, 5.0]$. The mean $\boldsymbol{\mu}$ is chosen as a point on the separatrix found by interpolating between two attractors and performing iterative binary search with ODE (1). A second $\boldsymbol{\mu}$ is obtained by interpolating a different attractor pair for training the second KEF.

**11D ecological dynamics (Fig. 6):** We similarly identify a single separatrix point $\boldsymbol{\mu}$. Each coordinate $x[i]$ is sampled independently from a Gamma distribution $x[i] \sim \Gamma(\alpha[i], \beta[i])$, where $\alpha[i]$ and $\beta[i]$ are chosen such that the mode of $x[i]$ equals $\mu[i]$ and the variance equals $\sigma_j^2$, with $\sigma_j = [0.01, 0.1, 0.3, 1.0]$.

**Data-trained RNN [9] (Fig. 8):** $\mathcal{N}\left(\boldsymbol{\mu}, \sigma_j^2 \tilde{\Sigma}\right)$, where we first construct a distribution which is oblongated along the direction of the attractors and isotropic along the remaining directions:

$$\Sigma = \sigma_B^2 \mathbf{I}_N + (\sigma_A^2 - \sigma_B^2) \mathbf{u} \mathbf{u}^\top,$$

with

- $\mathbf{u} \in \mathbb{R}^N$ a unit vector along the attractor axis, i.e. $\mathbf{u} = (\mathbf{a} - \mathbf{b})/\|\mathbf{a} - \mathbf{b}\|$, where $\mathbf{a}$ and $\mathbf{b}$ are the two attractors,
- $\sigma_A > 0$ the standard deviation along $\mathbf{u}$,
- $\sigma_B > 0$ the standard deviation along all orthogonal directions, and
- $\mathbf{I}_N$ the $N \times N$ identity matrix.

As before $\boldsymbol{\mu}$ is the point on the separatrix along the line joining the attractors.

We set $\sigma_A$ to include both attractors, and choose $\sigma_B$ as large as possible while avoiding spurious basins. For 300 samples drawn from $\mathcal{N}(\boldsymbol{\mu}, \Sigma)$, we evolve the dynamics forward for 3.0 s and estimate the covariance of the resulting approximately forward-invariant distribution, denoted $\tilde{\Sigma}$.

## H  Scaling of compute with Dimensionality

We ran all experiments on a system with four GeForce GTX 1080 GPUs with 10 Gbps of memory each.

All the 2D systems take 1-5 minutes to train the KEFs. The 11D gLV takes up to 20 minutes. The 668D data-trained RNN [9] takes 5 minutes to train the KEF.

Scalability to high-dimensions is a key strength of our approach. We evaluated the scaling behavior of our method on vanilla RNNs of varying sizes, each trained on the 1-bit flip-flop task. In table 2,

we report the wall-clock training time and curve-based validation performance for learning a single Koopman eigenfunction:

Table 2: Training time and performance across dimensionalities. Vanilla RNNs with $N$ units were trained on the 1-bit flip-flop task. We then applied our method to each resulting dynamical system $f$ to approximate its corresponding Koopman eigenfunction.

| Dimension $N$ | Wall-clock time (s) | Curve $R^2$ |
|---|---|---|
| 32 | 529 | 0.997 |
| 64 | 527 | 0.995 |
| 128 | 572 | 0.867 |
| 256 | 611 | 0.996 |
| 512 | 718 | 0.996 |

For each case, we verified good agreement with the ground truth using the curve-based validation metric (Fig. 8A–C). All models used a fixed DNN architecture with depth 20 and width 550. For larger $N$, we expect that the network width must scale with dimensionality. Since our method involves solving an $N$-dimensional PDE, its scaling behavior is expected to resemble that of Physics-Informed Neural Networks (PINNs) [26] and other PDE-solving neural methods [24]. In particular, Lu et al. [56] proved $N$-independent generalization error bounds for the Deep Ritz Method within a class of PDEs. Addressing the curse of dimensionality remains an active area of research [57, 58], and we anticipate that advances from this literature can be integrated into our framework.

# I  Choice of eigenvalue for numerics

In the main text we look for approximations to the Koopman PDE (3) for a real positive eigenvalue $\lambda$. What should the value of $\lambda$ be? It is known that products of KEFs are KEFs themselves with different eigenvalues. In particular, for a KEF $\psi$ with eigenvalue $\lambda$, we see that:

$$\nabla \left[ \psi(x)^\alpha \right] \cdot f(x) = \alpha \psi(x)^{\alpha-1} \nabla \psi(x) \cdot f(x) \tag{63}$$
$$= \alpha \lambda \psi(x)^\alpha \tag{64}$$

Therefore, $\psi(x)^\alpha$ is also a Koopman eigenfunction, with eigenvalue $\alpha\lambda$. This translates to changes in the shape of the KEF, i.e., the sharpness of the peaks, while maintaining the position of the zeroes.

In practice the choice of $\lambda$ affects training convergence, and it is therefore an important hyperparameter in the optimisation procedure (see Figure 10). We attribute this to the time scale of interest in the system $\dot{\boldsymbol{x}} = f(\boldsymbol{x})$, and differences in the propagation of gradients for different $\lambda$.

# J  Linking Koopman Eigenfunctions and separatrices: Formal Derivations

**Setting.**  Let $(\mathcal{X}, d)$ be a smooth manifold with metric $d$ and flow $\{\Phi^t\}_{t \in \mathbb{R}}$ generated by the autonomous ODE $\dot{\boldsymbol{x}} = f(\boldsymbol{x})$. Thus $\Phi^0 = \mathrm{id}$, $\Phi^{t+s} = \Phi^t \circ \Phi^s$, and $(t, \boldsymbol{x}) \mapsto \Phi^t(\boldsymbol{x})$ is continuous.

**Invariant set.**  An invariant set of a flow $\Phi^t$ is a subset $S \subset \mathcal{X}$ such that

$$\Phi^t(\boldsymbol{x}) \in S \quad \text{for all } \boldsymbol{x} \in S \text{ for all } t \in \mathbb{R}.$$

**Attracting set.**  A nonempty closed set $A \subset \mathcal{X}$ is an *attracting set* if it is invariant and there exists an open neighbourhood $U$ of $A$ such that

$$\lim_{t \to \infty} \mathrm{dist}(\Phi^t(\boldsymbol{x}), A) = 0 \quad \text{for all } \boldsymbol{x} \in U,$$

where for $\boldsymbol{y} \in \mathcal{X}$ and $A \subset \mathcal{X}$,
$$\mathrm{dist}(\boldsymbol{y}, A) := \inf_{\boldsymbol{a} \in A} d(\boldsymbol{y}, \boldsymbol{a}).$$

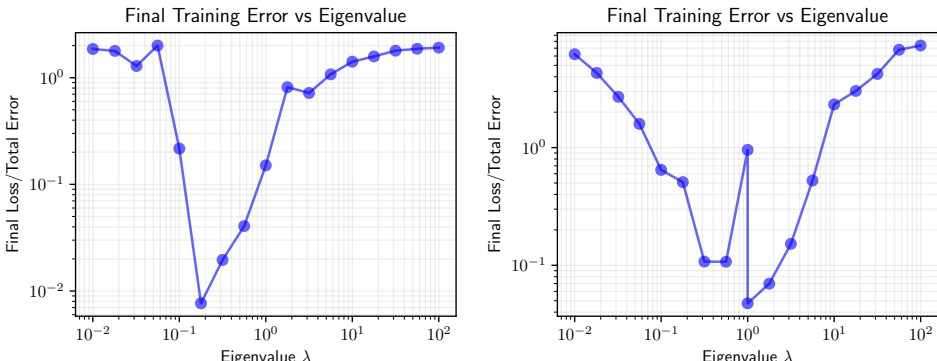

Figure 10: Training convergence as a function of eigenvalue $\lambda$, evaluated by normalised PDE error $\mathcal{L}_{\mathrm{ratio}}$ 6 for two systems: LEFT, 1D bistable system $\dot{x} = x - x^3$ (see Figure 3) and 2BFF GRU 3D (see Figure 5).

**Attractor.** An attracting set $A$ is an *attractor* if there is no proper subset of $A$ that is also an attracting set.

**Basins of attraction.** The basin of an attractor $A$ is

$$B(A) := \{\boldsymbol{x} \in \mathcal{X} : \lim_{t \to \infty} \mathrm{dist}(\Phi^t(\boldsymbol{x}), A) = 0\}.$$

Each $B(A)$ is forward invariant and open.

**Separatrix.** Let $\{A_k\}_{k \in K}$ be the set of all the attractors of $\Phi^t$ in $\mathcal{X}$. Define the separatrix as the complement of all basins:

$$\Sigma := \mathcal{X} \setminus \bigcup_{k \in K} B(A_k).$$

**Koopman eigenfunction.** Let $\psi : \mathcal{X} \setminus (\cup_k A_k) \to \mathbb{R}$ be continuous and satisfy

$$\psi\big(\Phi^t(\boldsymbol{x})\big) = e^{\lambda t} \psi(\boldsymbol{x}) \qquad \forall \boldsymbol{x} \in \mathcal{X} \setminus \bigcup_k A_k, \ \forall t \geq 0,$$

for some eigenvalue $\lambda > 0$. Then the sets $\{\psi > 0\}$, $\{\psi < 0\}$, and $\{\psi = 0\}$ are forward invariant.

**Constant sign near attractor (CS).** Let $\psi : \mathcal{X} \setminus (\cup_k A_k) \to \mathbb{R}$ be a continuous function. We say it has constant sign near an attractor $A_k$ if there exists an open neighbourhood $U_k$ with $A_k \subset U_k \subset B(A_k)$ such that $\psi$ has a constant sign on $U_k \setminus A_k$.

**Proposition 1** (Sign change at a zero $\Rightarrow$ separatrix point). *Let $\psi : \mathcal{X} \setminus (\cup_k A_k) \to \mathbb{R}$ be a continuous Koopman eigenfunction with positive eigenvalue $\lambda > 0$:*

$$\psi(\Phi^t(\boldsymbol{x})) = e^{\lambda t} \psi(\boldsymbol{x}), \qquad t \geq 0,$$

*and have constant sign near all the attractors $A_k$. Suppose $\boldsymbol{x} \in \mathcal{X} \setminus (\cup_k A_k)$ satisfies $\psi(\boldsymbol{x}) = 0$ and, for every $\varepsilon > 0$, the ball $B_d(\boldsymbol{x}, \varepsilon)$ contains points $\boldsymbol{y}^+, \boldsymbol{y}^-$ with $\psi(\boldsymbol{y}^+) > 0$ and $\psi(\boldsymbol{y}^-) < 0$. Then $\boldsymbol{x} \in \Sigma$.*

*Proof.* Assume for contradiction that $\boldsymbol{x} \in B(A_{k^*})$. Since $B(A_{k^*})$ is open, there exists $\varepsilon_0 > 0$ such that $B(\boldsymbol{x}, \varepsilon_0) \subset B(A_{k^*})$.

By (CS), $\psi$ has a constant sign on $U_{k^*} \setminus A_{k^*}$ for an open neighborhood $U_{k^*}$. By attractivity, for any $\boldsymbol{y} \in B(\boldsymbol{x}, \varepsilon_0)$ the trajectory $\Phi^t(\boldsymbol{y})$ eventually enters $U_{k^*}$ and remains there for all large $t$.

Since $\psi(\Phi^t(\boldsymbol{y})) = e^{\lambda t} \psi(\boldsymbol{y})$, the sign of $\psi(\boldsymbol{y})$ is preserved along the trajectory (forward invariance of $\{\psi > 0\}$ and $\{\psi < 0\}$), so $\psi(\boldsymbol{y})$ must already share that constant sign. Thus all points in $B(\boldsymbol{x}, \varepsilon_0)$ have the same sign, contradicting the assumption of sign change in every neighbourhood. Therefore $\boldsymbol{x} \notin \bigcup_k B(A_k)$, i.e. $\boldsymbol{x} \in \Sigma$. $\qquad\square$

**Remark.** The condition $\psi(\boldsymbol{x}) = 0$ with a sign change in every neighbourhood means that $\boldsymbol{x}$ is a boundary point of $\{\psi > 0\}$ and $\{\psi < 0\}$. Proposition 1 shows such points lie precisely on the separatrix.

