# OpenReview forum: "Finding separatrices of dynamical flows with Deep Koopman Eigenfunctions"
_NeurIPS.cc/2025/Conference — NeurIPS 2025 poster_

### Official Review · Reviewer_MwkQ · 2025-06-06

**Clarity:** 3
**Significance:** 3
**Originality:** 3
**Rating:** 4
**Confidence:** 3

**Summary:**

The authors propose a deep learning approach for identifying separatrices of dynamical flows. The method first trains a model to obtain Koopman eigenfunctions with real positive eigenvalues, and subsequently detects separatrices by leveraging their precisely vanishing characteristics at these boundaries. The effectiveness of the proposed approach is validated through multiple designed experiments.

**Questions:**

1. Has the author provided a comprehensive review of the relevant work on the discovery of separatrices? The article lacks comparative experiments with baselines, making it unclear whether the proposed method outperforms prior approaches.

2. The training of Koopman eigenfunctions requires global data near separatrices. Can the authors illustrate the training data conditions? Is it possible to directly estimate the separatrices from these trajectory data from training set? What are the advantages of the proposed method over direct estimation?

3. The paper lacks theoretical and experimental studies on the robustness of the proposed method. For example, how does it perform under complex scenarios such as noisy or partially observed data?

**Ethical Concerns:**

["NO or VERY MINOR ethics concerns only"]

**Final Justification:**

Although the article has certain limitations in comparative validation with baseline methods, the overall work still holds significant theoretical value in the field of nonlinear dynamics. Therefore, I am pleased to raise my score to 4.

**Limitations:**

Yes.

**Paper Formatting Concerns:**

No major formatting issues found. Paper adheres to NeurIPS 2025 guidelines.

**Quality:**

3

**Strengths And Weaknesses:**

**Strengths:**

1. The paper is logically structured and rigorously presented.

2. Extensive experimental validation has been conducted across multiple simulation systems.

3. The identification of separatrices holds significant application potential for the study of complex systems.

**Weaknesses:**

1. The paper lacks a detailed review of related work on separatrices discovery and comparative experiments with baseline methods.

2. Training Koopman eigenfunctions appears to require global data near separatrices, yet it remains unclear whether such trajectory data can be directly utilized for separatrices discovery.

3. The paper lacks both theoretical and experimental investigations into the robustness of the proposed method.

---

> ### Author Rebuttal · Authors · 2025-07-29
>
> # Response to Reviewer MwkQ
>
> We thank the reviewer for the helpful comments and constructive critique. Below, we address the main concerns raised regarding comparisons to prior work, the role of training data, and robustness.
>
> ---
>
> **1. Review of related work and comparisons with baselines**
>
> We agree that the paper would benefit from a broader discussion of existing methods for separatrix discovery. As also noted by Reviewer dWGz, one possible comparison is with approaches based on **Finite-Time Lyapunov Exponents (FTLE)**, which identify ridges of sensitivity in the flow. We now discuss FTLE and its differences from our method in our response to Reviewer dWGz. Briefly, FTLE does not identify zero-level sets, and its computation depends on simulation time $T$, while our approach yields a smooth, differentiable function whose zero-level set can be used directly to extract separatrices and optimize perturbations. We will revise the related work section to explicitly include this discussion.
>
> ---
>
> **2. Can the separatrix be directly estimated from trajectory data?**
>
> This is a very interesting question. Our current method assumes access to a continuous or inferred vector field $f(x)$ and learns Koopman eigenfunctions from it. We agree that directly estimating the separatrix from trajectory data — without modeling the flow — is an intriguing direction, though potentially difficult in practice. Separatrices are global structures and cannot generally be identified from local trajectory features alone.
>
> Thus, we view the two-step process (first infer $f$, then infer the separatrix) as a natural and principled strategy. In neuroscience, for instance, models of dynamics can be learned from neural recordings or trained task-specific networks (such as in our flip-flop examples). Both modeling paradigms — data-driven and task-driven — have yielded important insights into neural computation [2,3]. As we explain in the response to Reviewer dWGz ("**Neuroscience framing is unclear**"), these insights are thanks to methods in this spirit, that aid finding dynamical objects in high-dimensions [4].
>
> ---
>
> **3. Robustness under noise or partial observability**
>
> We appreciate this suggestion. While our current study focuses on fully observed deterministic systems, we believe the method can be extended to stochastic and partially observed settings. For example, the **gpSLDS** model [1] infers latent dynamics from noisy observations and provides uncertainty estimates over the learned flow field. Our method can be applied to such learned $f(x)$ functions directly, and the reported uncertainty in $f(x)$ can propagate to estimate uncertainty in the inferred separatrix.
>
> We mention this in our response to Reviewer dWGz and will add this future direction to the discussion section.
>
> ---
>
> **References**
>
> [1] Hu, Amber, et al. *Modeling latent neural dynamics with Gaussian process switching linear dynamical systems*. NeurIPS 2024
> [2] Vyas, Saurabh, et al. *Computation through neural population dynamics*. Annual Review of Neuroscience 43.1 (2020): 249–275
> [3] Barak, Omri. *Recurrent neural networks as versatile tools of neuroscience research*. Current Opinion in Neurobiology 46 (2017): 1–6
> [4] Sussillo, David, and Omri Barak. “Opening the black box: low-dimensional dynamics in high-dimensional recurrent neural networks.” _Neural computation_ 25.3 (2013): 626-649.

---

> > ### Comment · Reviewer_MwkQ · 2025-08-03
> >
> > After reviewing the authors' rebuttal, my concerns remain inadequately addressed.
> >
> > On one hand, if finding separatrices of dynamical flows is indeed a critical task, substantial prior work should exist, necessitating thorough comparisons with SOTA baseline methods.
> >
> > On the other hand, clear visualization of the training data used by the proposed method is essential—since the data itself could intuitively reveal separatrices.
> >
> > Given these unresolved issues, I maintain my original rating.

---

> > > ### Author Response · Authors · 2025-08-04
> > > **addressing lack of baseline methods, visualisation of training distributions**
> > >
> > > We thank the reviewer for following up and for raising two important points. We respond to each in turn below.
> > >
> > > ---
> > >
> > > **1. Lack of comparisons to baseline methods**
> > >
> > > We agree that benchmarking is essential for scientific progress. However, as noted in multiple recent papers [1–3], the field of computational neuroscience currently lacks standardized tasks and benchmarks for many analysis methods — particularly those focused on dynamical systems. The few benchmarks that do exist [1–3] primarily target latent variable inference or predictive modeling, rather than analysis tools like ours.
> > >
> > > Our work addresses a complementary challenge: identifying separatrices in systems where dynamics may be learned from data or trained on a task, in the spirit of [4]. That work introduced fixed-point analysis for trained RNNs, which has since proven highly influential. Yet, as with our approach, it was not benchmarked against baselines at the time — because no comparable methods existed, and its utility only became clear in retrospect [references 6-12 in submission]. We view our work similarly: identifying a new object of study (separatrices) that can yield insights into trained RNNs. Demonstrating its relevance and potential utility, as we do here, is itself a key contribution.
> > >
> > > We will add a comparison to FTLE (see response to dWGz) in the related work and discussion, which to our knowledge has not been used to design optimal perturbations as we demonstrated.
> > >
> > > ---
> > >
> > > **2. Visualization of training data**
> > >
> > > We completely agree that understanding the sampling distributions used during training is important, particularly to ensure that the separatrix is not trivially revealed by the data distribution. In our method, we sample points $x$ from a chosen distribution, evaluate $f(x)$, and train the Koopman eigenfunction $\psi(x)$ to satisfy the PDE constraint $\nabla \psi(x) \cdot f(x) = \lambda \psi(x)$. The choice of sampling distribution directly influences where the PDE is enforced.
> > >
> > > To clarify this aspect, we will include visualizations of the training distributions for all examples in the revised appendix. This will help ensure transparency and allow readers to assess whether the method relies on trivial cues from the sampling strategy.
> > >
> > > ---
> > >
> > > **References**
> > >
> > > [1] Pei, Felix, et al. *Neural latents benchmark'21: evaluating latent variable models of neural population activity.* arXiv:2109.04463 (2021).
> > > [2] Versteeg, Christopher, et al. *Computation-through-Dynamics Benchmark: Simulated datasets and quality metrics for dynamical models of neural activity.* bioRxiv (2025).
> > > [3] Schrimpf, Martin, et al. *Brain-score: Which artificial neural network for object recognition is most brain-like?* bioRxiv:407007 (2018).
> > > [4] Sussillo, David, and Omri Barak. *Opening the black box: low-dimensional dynamics in high-dimensional recurrent neural networks.* Neural Computation 25.3 (2013): 626–649.

---

### Official Review · Reviewer_dWGz · 2025-06-12

**Clarity:** 4
**Significance:** 2
**Originality:** 2
**Rating:** 5
**Confidence:** 4

**Summary:**

The paper presents a new approach to identify and characterize boundaries between basins of attraction (separatrices). It uses Koopman Eigenfunctions (KEFs) with positive eigenvalues that vanish near separatrices. The method locates points in state space where these positive KEFs equal zero to identify separatrices. The authors demonstrate the approach on multiple datasets, including synthetic data, ecological models, and recurrent neural networks. They also show the method’s potential by finding optimal perturbations to shift systems between basins.

**Questions:**

1) Regarding “encouraging sign changes across the separatrix” (line 136): I don’t understand how the cost in Eq. 10 forces a sign change. I see how it promotes non-zero values on both sides of the separatrix, but it seems possible for the eigenfunction to have the same sign on both sides while passing through zero at the separatrix. Could you please clarify this?

2) About the steps around Eqs. 13 and 14: Do you first find individual separatrices and then combine them? In other words, do you solve Eq. 14 for each $\psi$ separately and then combine the separators afterward? Why did you choose to present Eq. 13 before Eq. 14? I think reversing this order would improve the flow.


3) In Eqs. 15–16, why do you assume Euclidean distance? Could the relevant effort be in a different metric space instead?

4) Why did you choose to use a DNN to approximate the Koopman eigenfunctions instead of directly optimizing the loss functions (e.g., via EM or variational methods) without a network?

5) It seems that for most examples, the authors included some ground truth knowledge that helped with evaluation. While I agree it makes sense to evaluate the method using examples where we have some idea of what the separatrices should look like, I wonder how would you evaluate the method in the absence of ground truth data? That is, are there any properties of either the separatrix or the values of the eigenfunctions around the separatrix that could help assess the certainty of the identified boundary when ground truth is unavailable?

**Ethical Concerns:**

["NO or VERY MINOR ethics concerns only"]

**Final Justification:**

I thank the authors for their response and believe that the clarifications, additional comparisons, and discussion points will strengthen the paper.

Following point 1, one more suggestion for the discussion: I wonder if, when applying models like rSLDS, the separatrices might be captured by the switching regions between the LDSs, or if residual errors in model fitting could slightly distort their locations, making them harder to detect. It would be interesting to compare this (discussion-wise, as I understand more experiments are beyond the scope) to what more continuous models (e.g., those I mentioned in my review) might capture.

I have raised my score to 5.

**Limitations:**

Yes, also see in previous cell under Weaknesses

**Quality:**

4

**Strengths And Weaknesses:**

I enjoyed reading the paper and believe it addresses important challenges in dynamical systems modeling.

## Strengths:
1) The paper is very well-written, with the motivation and approach clearly explained and presented in a way that is easy to follow.
2) Figures are visually appealing and reflect the strengths of the approach.
3) The examples the authors chose clearly demonstrate varying aspects and strengths of the method, and encompass different dimensionalities and stability properties (e.g., fixed points vs. limit cycles) across different application domains.




## Weaknesses:
### **Major:**
1) The method feels neither fully theoretical nor purely data-driven (i.e., trajectories data without knowing the dynamics in the full phase space), but more conceptual. Since it sits somewhere in between, I think it’s important to highlight how it can actually help science in practice given data. You touch on this a bit in the discussion, but I’d appreciate one or two more specific sentences explaining how it could be applied, even in future work, using data or combined with data-driven approaches. For example, maybe first applying SINDy [1] to identify basis dynamic “atoms” to better understand the flow field, or using discrete data-driven models like SLDS [2,3] / dLDS [4,5] to identify transitions that then help infer the continuous flow field, which could then be used with your method. Notably, I do not suggest applying these approaches here, as that is beyond the scope of the paper, but rather recommend including a discussion on how your method could be combined with such data-driven techniques to characterize dynamics from data.

2) I do not understand how the cost in Eq. (10) forces a sign change. It seems like it promotes non-zero values on both sides of the separatrix, but the eigenfunction could still have the same sign on both sides and simply pass through zero on the separatrix. Can you clarify this?

3) I miss any comparisons to existing methods. While I understand that there are not many approaches that target idnetifying separatrices, some comparison to current tools (e.g. FTLE) is important.

4) While the cost terms are explained, I feel the practical step-by-step process of the method could be clearer via an algorithm or pseudocode (e.g., using the algorithm package in LaTeX).


### **Minor:**
1) Line 95 — please refer to the specific appendix (seems like Appendix C?).
2) Typos: 1) Line 124:  “We” should not be capitalized. 2) Line 168: extra “)”. 3) Line 197: “it’s” -> “its”.
3) Throughout the paper, I felt the authors tried to connect their method to neuroscience, but I think the connection should either be more clearly emphasized or removed, as right now it feels somewhat in between and unclear (parsing-wise). I suggest either focusing the paper solely on dynamical systems understanding, perhaps mentioning neuroscience as just one example among others, or, alternatively, strengthening the neuroscience link. For example, the authors could emphasize more clearly why the RNN task is actually neuroscience-relevant, or discuss the neuro significance in greater detail,like how it can be helpful in the future for BCI interventions, or  for designing brain perturbations for motor control, etc.
4) I would be more cautious when discussing perturbations. The method seems to assume strong conditions about the effect of the perturbation, for example, that it does not alter the state space or the separatrices themselves, which may not hold in all systems. It is important to clearly state such assumptions about the nature of the perturbations.
5) In Eqs. (15–16), I do not think the Euclidean distance is necessarily the best measure for "minimal perturbations." I think it would be helpful if you could provide more details about how you assume the perturbation occurs. For example, are they operators that simply move the system to a new state directly, or do they involve pushing the system against the flow field? If the former, why should the distance matter? If the latter, if the perturbation requires pushing the system to a new location in an energy-consuming way, it might be better to consider the flow field’s density and magnitude throughout the perturbation. Geometrical distance may not correlate with the actual effort needed to push against the flow. Providing a specific example of the type of perturbation you have in mind and explaining why would help. You could also mention that energy-consumption considerations might be added in future work and justify the current use of Euclidean distance for computational simplicity.
6) I missed a colorbar for the background in all phase portraits. Even if the exact values don’t matter, having a colorbar in at least one of the figures (and indicating the direction—e.g., blue for 0, white for high values) would be helpful for interpretation.



[1] Brunton, S. L., Proctor, J. L., & Kutz, J. N. (2016). Discovering governing equations from data by sparse identification of nonlinear dynamical systems. Proceedings of the national academy of sciences, 113(15), 3932-3937.

[2] Linderman, S. W., Miller, A. C., Adams, R. P., Blei, D. M., Paninski, L., & Johnson, M. J. (2016). Recurrent switching linear dynamical systems. arXiv preprint arXiv:1610.08466.

[3] Linderman, S., Johnson, M., Miller, A., Adams, R., Blei, D., & Paninski, L. (2017, April). Bayesian learning and inference in recurrent switching linear dynamical systems. In Artificial intelligence and statistics (pp. 914-922). PMLR.

[4] Mudrik, N., Chen, Y., Yezerets, E., Rozell, C. J., & Charles, A. S. (2024). Decomposed linear dynamical systems (dlds) for learning the latent components of neural dynamics. Journal of Machine Learning Research, 25(59), 1-44.

[5] Chen, Y., Mudrik, N., Johnsen, K. A., Alagapan, S., Charles, A. S., & Rozell, C. (2024). Probabilistic decomposed linear dynamical systems for robust discovery of latent neural dynamics. Advances in Neural Information Processing Systems, 37, 104443-104470.

---

> ### Author Rebuttal · Authors · 2025-07-29
>
> # Response to Reviewer dWGz
>
> We thank the reviewer for the thoughtful and constructive feedback. We are especially grateful for the positive assessment of the paper’s clarity, examples, and overall presentation. Below, we respond to the specific concerns raised.
>
> ---
>
> ## Major Weaknesses
>
> **1. Application to real neural data / data-driven models**
>
> We agree this is a critical aspect. Our method can be applied to any model that infers a flow field $f(x)$ from data. This includes GPSDE [1], rSLDS [2], gpSLDS [3].
>
> These are all stochastic models, but one can apply our method considering a deterministic component or approximation of the inferred dynamics. We developed this idea in the case of rSLDS and GPSDE as they have been widely applied to neural data [9]. (See our response to reviewer FEbm). We tested our method on rSLDS trained on data from a synthetic bistable stochastic system (response to reviewer FEbm). In gpSLDS [3], uncertainty in the flow is also reported; this can be propagated to estimate the uncertainty in the separatrix.
>
> We thank you for the suggestion regarding rSLDS and believe the result strengthens our work and would be happy to include it. We will expand our discussion to explicitly describe how our approach complements these data-driven models and can be applied post hoc to real data.
>
> **2. Clarification on Eq. (10): how does it promote sign changes?**
>
> Thank you for catching this. The loss term in Eq. (10) not only promotes non-zero values but also encourages the eigenfunction $\psi$ to have both positive and negative values across a region—thus promoting sign changes. This is achieved by minimizing the squared mean relative to the variance:
>
> $$
> \frac{\langle \psi \rangle^2}{\langle \psi^2 \rangle} \ll 1
> $$
>
> When the mean is close to zero but the variance is high, $\psi$ necessarily has values of opposite sign. We will clarify this in the text and add annotations to Figure 3 to highlight this behavior.
>
> **3. Comparison to existing methods like Finite Time Lyapunov Exponent (FTLE)**
>
> Thank you for bringing our attention to this work. FTLE methods characterise regions of sensitivity via divergence rates of trajectories. Ridges of the FTLE correspond to separatrices.
>
> We draw some comparisons to our method.
>
> While our method yields a function which is zero along the entire separatrix, the FTLE takes non-zero positive value that may vary along the separatrix. This makes it more difficult to determine if a point is a separatrix or not.
> Our KEF-based method yields a *differentiable* function whose zero set explicitly defines a boundary; this enables efficient gradient-based search for minimal perturbations. Differentiating FTLE requires differentiating through the dynamical function which may not be possible (vanishing gradients) or may be computationally expensive.
> FTLE requires forward simulation for a time $T$, while our method avoids this step and may offer improved computational efficiency once the eigenfunction is trained.
>
> We will include a brief comparison with FTLE in the related work section.
>
> **4. Include an explicit algorithm or pseudocode**
>
> We agree this would improve clarity. We will include pseudocode using the `algorithm` package in LaTeX to summarize our method step-by-step in the revised version.
>
> ---
>
> ## Minor Issues
>
> **1. Line 95 — reference to specific appendix**
>
> We will update this to explicitly refer to Appendix C.
>
> **2. Typos**
> We thank the reviewer for pointing them out. We will make these corrections.
>
> **3. Neuroscience framing is unclear**
>
> We thank the reviewer for this valuable advice. We can better highlight the relevance of our method to neuroscience. While our discussions address *data-trained* dynamical models of the brain, such models may also be *task-trained*. In the past, both types have generated insights about neural computations (see [4,5] for reviews). In particular the task-trained variants serve as hypothesis generation tools for how a certain computation could be carried out by a recurrent neural circuit [5]. Such models often posses dynamical objects like fixed-points, line-attractors, oscillations, and separatrices but are embedded in a high-dimensional space making it harder to identify. This motivates methods for finding/studying these objects [6]. The flip-flop is a prototypical task yielding multi-stable dynamics in RNNs, and is often used to demonstrate dynamical tools in neuroscience [7,8]. But separatrices are more general and appear in almost any scenario involving discrete choices, memories, threshold behavior [9].
>
> We will add this to our introduction and discussion.
>
>
> **4. Assumptions about perturbations part 1**
> Thank you, we did not clearly specify whether the input perturbations affect the recurrent dynamics, i.e., $f(x)$ versus $f(x,u)$ where u is an external input, e.g., a stimulus or neural perturbation.  In the paper we assumed that it does not $f(x)$. However, our formalism can be extended to $f(x,u)$ by defining $\psi(x,u)$, in which case the Koopman PDE becomes
> $$\nabla_x \psi(x,u)\cdot f(x,u)=\psi(x,u).$$ We leave the development of this scenario to future work.
>
> We will specify our assumption that the flow is independent of $u$ in our paper.
>
>
> **5. Assumptions about perturbations part 2**
>
> We apologise, we did not clearly explain our perturbation model.
> In the paper, we consider an idealized perturbation: a powerful Dirac delta pulse that instantaneously pushes the system from its current state to a new one, i.e., $x_\text{base} + \delta x$. Under this model, the optimal perturbation is simply the one with minimal norm, $|\delta x|$, subject to $\psi(x_\text{base} + \delta x) = 0$ — as shown in Eq. (15).
>
> A more general version of this problem would involve finding a time-varying input signal that drives the system across the separatrix while minimizing energy or power. This still requires identifying a final state $x_\text{final}$ such that $\psi(x_\text{final}) = 0$, and minimizing the cost of reaching it (e.g., via a control integral along the trajectory from $x_\text{init}$ to $x_\text{final}$).
>
> Our current focus is on identifying the separatrix via $\psi(x)$ — not on solving the control problem itself. However, the use of a smooth, differentiable eigenfunction to define the boundary is compatible with broader control formulations, and could serve as a basis for future work incorporating energy-aware perturbation metrics.
>
> We will add these details to the main text.
>
> **6. Missing colorbars in phase portraits**
>
> We thank the reviewer for this important point. We will add colorbars to the phase plots and clarify the meaning of background colors (e.g., blue = 0, white = high) to improve interpretability.
>
> ---
>
> ## Responses to Specific Questions
>
> **1. Clarify Eq. (10) and sign changes**
>
> See response under Major Weakness #2.
>
> **2. Order of Eqs. (13) and (14)**
>
> You are right — we should present Eq. (14) before Eq. (13), since we solve for individual KEFs first and then combine them. We will revise the order and clarify in the text. For example, in Figure 3, the two KEFs are shown separately in Figure 4.
>
> **3. Why use Euclidean distance for perturbations?**
>
> See response under Minor Issue #5. In brief: we assume a direct displacement via a delta-pulse perturbation. In that case, Euclidean norm is appropriate. Future work could incorporate flow-aware or energy-based metrics.
>
> **4. Why use DNNs for KEFs instead of EM or variational methods?**
>
> We use DNNs as a flexible function approximator to parameterize $\psi$ and optimize the PDE residual. DNNs are expressive and differentiable, allowing us to not only solve the PDE but also optimize over $\psi(x)$ post-training. While one could in principle use other parameterizations, we are not aware of EM formulations that apply here, since our setup does not involve a probabilistic latent variable model. If the reviewer had a specific alternative in mind, we would be happy to consider it.
>
> **5. Evaluation when no ground truth is available**
>
> In systems without known separatrices, we can validate a candidate point $x$ on the predicted separatrix (i.e., $\psi(x) = 0$) by evolving nearby initial conditions. If they flow to different attractors, this supports the claim that $x$ lies on a separatrix. This is related to FTLE-type validation.
>
> We will include a discussion of this strategy in the final version.
>
> ---
>
> ## References
>
> [1] Duncker et al., *Learning interpretable continuous-time models of latent stochastic dynamical systems*, ICML 2019
> [2] Linderman et al., *Bayesian learning and inference in recurrent switching linear dynamical systems*, AISTATS 2017
> [3] Hu et al., *Modeling latent neural dynamics with Gaussian process switching linear dynamical systems*, NeurIPS 2024
> [4] Vyas, Saurabh, et al. "Computation through neural population dynamics." Annual review of neuroscience 43.1 (2020): 249-275.
> [5] Barak, Omri. "Recurrent neural networks as versatile tools of neuroscience research." Current opinion in neurobiology 46 (2017): 1-6.
> [6] Sussillo, David, and Omri Barak. "Opening the black box: low-dimensional dynamics in high-dimensional recurrent neural networks." _Neural computation_ 25.3 (2013): 626-649.
> [7] Versteeg, Christopher, et al. "Computation-through-Dynamics Benchmark: Simulated datasets and quality metrics for dynamical models of neural activity." _bioRxiv_ (2025)
> [8] Ostrow, Mitchell, et al. "Beyond geometry: Comparing the temporal structure of computation in neural circuits with dynamical similarity analysis." _Advances in Neural Information Processing Systems_ 36 (2023): 33824-33837.
> [9] Nair, Aditya, et al. "An approximate line attractor in the hypothalamus encodes an aggressive state." _Cell_ 186.1 (2023): 178-193.

---

### Official Review · Reviewer_FEbm · 2025-07-01

**Clarity:** 3
**Significance:** 2
**Originality:** 3
**Rating:** 4
**Confidence:** 2

**Summary:**

This paper presents an innovative deep learning framework for identifying separatrices in high-dimensional dynamical systems using Koopman eigenfunctions (KEFs). The authors develop a neural network-based approach to approximate KEFs with positive eigenvalues, which naturally vanish on separatrices, enabling efficient boundary detection through gradient-based optimization. The method demonstrates strong empirical performance across diverse systems including synthetic benchmarks, ecological networks, and recurrent neural networks, while also providing practical applications for designing optimal perturbations in neuroscience contexts.

**Questions:**

How does the method's performance scale with increasing system dimensionality, particularly beyond the 64D case demonstrated?

Could you elaborate on the theoretical connection between the zero-level sets of KEFs and true separatrices in more complex, non-analytical systems?

What modifications would be needed to apply this approach to stochastic dynamical systems where separatrices may not be sharply defined?

Have you explored alternative neural architectures beyond RBF networks that might improve approximation accuracy or training efficiency?

**Ethical Concerns:**

["NO or VERY MINOR ethics concerns only"]

**Final Justification:**

The method offers a fresh perspective on separatrix learning, but the lack of rigorous guarantees in high dimensions and stochastic environments, along with scalability concerns, restricts its broader applicability.

**Limitations:**

Yes

**Quality:**

3

**Strengths And Weaknesses:**

The methodology shows promising results across diverse systems (from 1D synthetic models to 64D RNNs), its theoretical foundations remain underdeveloped, lacking rigorous guarantees linking KEF approximation accuracy to separatrix identification. The approach creatively addresses degeneracy challenges through balance regularization and ensemble training, as illustrated in Figure 3, but computational scalability for very high-dimensional systems may be limited by the iterative PDE-solving process. The work makes significant contributions to dynamical systems analysis by bridging Koopman theory with practical neural network applications, though broader impacts on real-world neuroscience interventions require further validation.

---

> ### Author Rebuttal · Authors · 2025-07-29
>
> # Response to Reviewer FEbm
>
> We thank the reviewer for the thoughtful and constructive comments. Below, we address the main concerns and questions raised.
>
> ---
>
> **Broader impacts on neuroscience interventions**
>
> We agree that experimental validation is ultimately required to assess real-world impact. However, as we discussed in our response to Reviewer dWGz, our method can be applied to *data-derived models*, including those learned from neural data. This opens the door to *in silico* experimentation and the design of optimal perturbations, which can then be validated through targeted neuroscience experiments.
>
> ---
>
> **Scaling with dimensionality**
>
> Thank you for raising this point. Scalability to high dimensions is an important strength of our work. We tested scalability on the 1-bit flip-flop system (with $N = 64$ in the main text, Figure 7) in increasingly higher dimensions. On a single NVIDIA GeForce GTX 1080 Ti GPU, we measured the runtime for training a single Koopman eigenfunction:
>
> - $N = 32$ → 529 seconds
> - $N = 64$ → 527 seconds
> - $N = 128$ → 572 seconds
> - $N = 256$ → 611 seconds
> - $N = 512$ → 718 seconds
>
> For each, we verified a good match to ground-truth using the curve based validation (Fig 7A,B,C). We used a fixed DNN with depth = 20 and width = 550. For larger dimensions, we expect that width will need to scale with $N$. Since our method involves solving an $N$-dimensional PDE, its scaling behavior is similar to that of Physics-Informed Neural Networks (PINNs) and Deep Ritz methods, and addressing the curse of dimensionality is an active area of research [2,3]. We anticipate that advances from this literature can be integrated into our framework.
>
> ---
>
> **Theoretical connection between KEFs and separatrices**
>
> Prior work [5] has shown that for linear and globally stable nonlinear systems, there exist Koopman eigenfunctions whose level sets align with stable manifolds (i.e., separatrices). Our setting is more general: we make no assumptions about fixed points or global stability and learn eigenfunctions purely from the dynamics.
>
> As such, there is no strict one-to-one correspondence between the zero level set of a learned KEF and a true separatrix. In fact, we illustrate this explicitly in Figure 3:
> 1. A zero level set not aligned with a separatrix (top row, right)
> 2. A separatrix without a vanishing KEF (bottom row, $\mu = 0$ and $\mu = 1$)
>
> These examples are not pathologies, but expected outcomes under our relaxed assumptions. We analyze and address them using our degeneracy mitigation strategies (Figure 3, Section 3.2).
>
> ---
>
> **Application to stochastic dynamical systems**
>
> For stochastic systems of the form  $$
> dX = f(X)dt + \sigma(X)dW,
> $$  a common approach in the literature is to analyze the deterministic flow $f(X)$ by ignoring the noise term ($\sigma\equiv0$). This is frequently done in phase-plane and stability analysis [1]. Our method can be applied in this manner, with no modifications needed.
>
> In learned models, uncertainty in $f(X)$ due to limited data or inference procedures can be propagated to quantify uncertainty in the separatrix location [5]. However, this is beyond the scope of the current work.
>
> We also consider another class of stochastic models commonly applied to neural data: Recurrent Switching Linear Dynamics Systems (rSLDS)[6]. These consist of continuous and discrete stochastic variables coupled to each other. This yields regimes of the continuous phase space that are governed by different linear dynamics, probabilistically. To apply our method to such a model we first construct a smooth, deterministic approximation to the dynamics (details in appendix below), and then apply our method to it to obtain the KEF. We tested this idea on a 2D rSLDS with a 5-state discrete variable, trained on 2D synthetic data from a stochastic bistable system, then trained a KEF on the smoothed dynamics of the rSLDS and found the separatrix reliably. We would be happy to include our results in the final version to demonstrate the applicability to stochastic models and models derived from data.
>
>
> ---
>
> **Alternative architectures**
>
> The Radial Basis Function (RBF) network was used only for the 2D limit-cycle system (Figure 6). All other experiments used deep ResNets with tanh activation, as described in Appendix F. We apologize for the lack of clarity on this point in the main text and will revise accordingly.
>
> Our (unreported) experiments showed that ResNets outperform vanilla MLPs of comparable size in terms of stability and approximation accuracy. We also found tanh activations to be more stable than ReLU in our context.
>
> ---
>
> **On theoretical rigor**
>
> We acknowledge the absence of formal guarantees linking KEF approximation error to separatrix accuracy. While our empirical results are strong and supported by theoretical intuition (e.g., zero-level sets of certain KEFs align with separatrices), establishing tighter theoretical bounds remains an important direction for future work.
>
> ---
>
> **References**
>
> [1] Duncker et al., *Learning Interpretable Continuous-Time Models of Latent Stochastic Dynamical Systems*, ICML 2019
> [2] Hu et al., *Tackling the Curse of Dimensionality with Physics-Informed Neural Networks*, Neural Networks 2024
> [3] Lu et al., *A Priori Generalization Analysis of the Deep Ritz Method*, COLT 2021
> [4] Hu et al., *Modeling Latent Neural Dynamics with GP-SLDS*, NeurIPS 2024
> [5] Mezić, I., *Spectrum of the Koopman Operator, Spectral Expansions in Functional Spaces, and State-Space Geometry*, Journal of Nonlinear Science, 30(5), 2091–2145 (2020)
> [6] Linderman et al., Bayesian learning and inference in recurrent switching linear dynamical systems, AISTATS 2017
>
> ---
> **Appendix: Non-linear deterministic approximation of an rSLDS**
>
> To derive a smooth deterministic flow approximation of the rSLDS dynamics we consider the posterior $p(z|x)$ relating the discrete variable $z\in\{1,2,...,K\}$ -- which chooses the recurrent dynamics matrix $A_z\in \mathbb R^{N\times N}$ -- and the continuous variable $x\in \mathbb R^N$, whose dynamics are governed by $A_z$. One possibility for the flow is to choose the most-likely dynamics at each point,
> $$
> f(x) := A_{\tilde z(x)} x \quad \text{such that} \quad \tilde z(x) = \underset{z}{\text{arg max}} \ p(z \mid x).
> $$ This yields a piecewise linear function that may suffer from discontinuities. Another option is to take the 'expected' dynamics:
> $$f(x):=\langle A_z x\rangle_{z|x}=\sum_z p(z|x)A_{z}x.$$ This yields a smooth deterministic non-linear dynamical system. We can also choose an intermediate option by defining an inverse-temperature parameter $\beta$ and applying a softmax:
> $$p(z|x)\leftarrow \frac{e^{\beta \log p(z|x)}}{\sum_{z'}e^{\beta \log p(z'|x)}},$$ where $\beta\to\infty$ recovers the argmax version. In our numerical experiment we choose $\beta=2$, maintaining a close match to the original piece-wise linear system but smoothing the transitions between linear regimes.

---

> > ### Author Response · Authors · 2025-08-05
> > **follow up**
> >
> > Since we did not receive a response to our rebuttal, we are following up to check if there are any concerns remaining that we could address.

---

> > ### Comment · Reviewer_FEbm · 2025-08-07
> >
> > Thank you for the clarifications. Please make sure the final version (1) adds the scaling comparison table to the main paper (not just the rebuttal), (2) discusses the limits of sparsity and rapid graph changes more explicitly, and (3) briefly comments on the discrete-time / hybrid extension in the limitations section.

---

> > > ### Author Response · Authors · 2025-08-07
> > > **thanks and clarifications**
> > >
> > > We thank the reviewer for their response.
> > >
> > > We will include the results on scaling with dimensionality in the final version.
> > >
> > > Regarding points (2) and (3), we believe there may be some confusion. The terms sparsity, graph, and hybrid do not appear in our paper or in any prior discussion. As for discrete-time, our analysis is consistently presented in the continuous-time setting throughout the manuscript.
> > >
> > > We would greatly appreciate clarification on these points to ensure we are addressing the reviewer’s concerns appropriately.

---

### Author Response · Authors · 2025-08-09
**General Response**

We thank all the reviewers for their detailed feedback and valuable ideas.

Reviewers **MwkQ** and **dWGz** emphasized the importance of the separatrix-finding problem and commended the logical structure, clarity of writing, and breadth of our experimental validation. Reviewer **FEbm** described our approach as innovative, noted its strong empirical performance across diverse systems, and highlighted our creative handling of degeneracy challenges.

---
Below are the main concerns raised and how we address them:

**Theoretical foundation and performance guarantees**
We will expand the related work to include prior theoretical results establishing the existence of Koopman eigenfunctions with the desired properties, while noting that our setting of multistable systems is more general. We will clarify the (partial) guarantees that hold in known cases and outline open challenges for the general case.

**Comparison to existing methods**
Reviewer **dWGz** acknowledged that there are not many existing methods but drew our attention to Finite-Time Lyapunov Exponents (FTLE), which involves running the dynamics over a finite time and evaluating sensitivity to initial conditions. While comparing FTLE directly against our Koopman-based method is beyond the scope of this work, we draw clear comparisons in our response to **dWGz**, highlighting the relative strengths of our method. We will incorporate these into the related work and discussion in the final version.

**Practical applicability and data-driven scenarios**
We will add examples and discussion on how our method can be combined with data-driven modeling pipelines to operate when the full system equations are unknown, clarifying its applicability to real-world experimental data.

---
We appreciate that Reviewer **dWGz** recommends clear acceptance, and we thank all reviewers for their contributions, which we believe will further strengthen the final version of the paper.

---

### Decision · Program_Chairs · 2025-09-17

**Decision:**

Accept (poster)

**Comment:**

The authors of this work devised a method for tracing out separatrices in dynamical systems, the boundaries that separate different basins of attraction, based on Koopman operators with their eigenfunctions approximated by DNNs. They illustrate their approach on several low-dimensional examples, but also on two simulated high-dimensional systems, one from ecology and one an RNN trained on a one-bit flip-flop task.

This is an important topic, of considerable theoretical and practical interest across diverse fields dealing with complex, multi-stable dynamical systems, which hasn’t received that much attention so far. The authors are mindful of different failure modes (degeneracies) of their method, and provide sound solution strategies. Initially, reviews brought up a couple of issues regarding comparisons to existing methods, the broader applicability of the method, potential robustness issues, and potential scaling problems. In my mind these were mostly addressed by the authors. There are indeed not that many methods out there for detecting these manifolds, although there is the class of numerical continuation methods which the authors may have missed (and might want to include in their discussion). An application I would have found really interesting (also discussed by the referees) is on RNNs actually trained on real-world, e.g. neuroscience, data, not just simulated or artificial tasks. This could have made a much stronger point for the methods' virtues.

Overall, however, I find this a well-crafted and important contribution, with a solid technical part and several nice ideas along the way. I support acceptance, as did all three referees following the AC-referee discussion.